# Discovery, recognized antigenic structures, and evolution of cross-serotype broadly neutralizing antibodies from porcine B-cell repertoires against foot-and-mouth disease virus

**Fengjuan Li[1,2☯], Shanquan Wu[3☯], Lv Lv[1,2], Shulun Huang[1,2], Zelin Zhang[3], Zhaxi Zerang[3], Pinghua Li[1,2], Yimei Cao[1,2], Huifang Bao[1,2], Pu Sun[1,2], Xingwen Bai[1,2], Yong He[4], Yuanfang Fu[1,2], Hong Yuan[1,2], Xueqing Ma[1,2], Zhixun Zhao[1,2], Jing Zhang[1,2], Jian Wang[1,2], Tao Wang[1,2], Dong Li[1,2], Qiang Zhang[1,2], Jijun He[1,2], Zaixin Liu[1,2]\*, Zengjun Lu[1,2]\*, Dongsheng Lei[1,3]\*, Kun Li[1,2]\***

1 State Key Laboratory for Animal Disease Control and Prevention, College of Veterinary Medicine, Lanzhou University, Lanzhou Veterinary Research Institute, Chinese Academy of Agricultural Sciences, Lanzhou, P. R. China, 2 Gansu Province Research Center for Basic Disciplines of Pathogen Biology, Lanzhou, P. R. China, 3 School of Physical Science and Technology, Electron Microscopy Centre of Lanzhou University, Lanzhou University, Lanzhou, P. R. China, 4 School of Pharmaceutical Sciences, Shandong University, Ji'nan, P. R. China

☯ These authors contributed equally to this work.
\* liuzaixin@caas.cn (ZL); luzengjun@caas.cn (ZeL); leids@lzu.edu.cn (DL); likun02@caas.cn (KL)

**Data Availability Statement:** The single B cell sequencing (scBCR-seq and scRNA-seq) data generated in this study have been deposited into

## Abstract

It is a great challenge to isolate the broadly neutralizing antibodies (bnAbs) against foot-and-mouth disease virus (FMDV) due to its existence as seven distinct serotypes without cross-protection. Here, by vaccination of pig with FMDV serotypes O and A whole virus antigens, we obtained 10 bnAbs against serotypes O, A and/or Asia1 by dissecting 216 common clonotypes of two serotypes O and A specific porcine B-cell receptor (BCR) gene repertoires containing total 12720 B cell clones, indicating the induction of cross-serotype bnAbs after sequential vaccination with serotypes O and A antigens. The majority of porcine bnAbs (9/10) were derived from terminally differentiated B cells of different clonal lineages, which convergently targeted the conserved "RGDL" motif on structural protein VP1 of FMDV by mimicking receptor recognition to inhibit viral attachment to cells. Cryo-EM complex structures revealed that the other bnAb pOA-2 specifically targets a novel inter-pentamer antigen structure surrounding the viral three-fold axis, with a highly conserved determinant at residue 68 on VP2. This unique binding pattern enabled cross-serotype neutralization by destabilizing the viral particle. The evolutionary analysis of pOA-2 demonstrated its origin from an intermediate B-cell, emphasizing the crucial role of somatic hypermutations (SHMs) in balancing the breadth and potency of neutralization. However, excessive SHMs may deviate from the trajectory of broad neutralization. This study provides a strategy to uncover bnAbs against highly mutable pathogens and the cross-serotype antigenic structures to explore broadly protective FMDV vaccine.

the Sequence Read Archive database with the accession code of PRJNA1064841. The cryo-EM density maps and structures for FMDV-O18-pOA2 and FMDV-AWH-pOA2 have been deposited into the Electron Microscopy Data Bank (EMDB) and the Protein Data Bank (PDB) with the following accession numbers: FMDV-O18-pOA2, EMD-38814, PDB 8Y0Q, and FMDV-AWH-pOA2, EMD-38815, PDB 8Y0R. The sequences of all the porcine-derived monoclonal antibodies have been shown in S2 Table. All other relevant data are within the paper and its supporting information files.

**Funding:** This work was supported by grants from the National Key R&D Program of China (2021YFD1800304 to Z.L.), the National Natural Science Foundation of China (Nos. 32373028 to K.L., 32171300 to D.L., 32072873 to Y.C., and 32302884 to H.Y), Fundamental Research Funds for the Central Universities (lzujbky-2021-ct05 to D.L). The funders had no role in study design, data collection and analysis, decision to publish, or preparation of the manuscript.

**Competing interests:** The authors have declared that no competing interests exist.

## Author summary

The scarcity of broadly neutralizing antibodies (bnAbs) in the body significantly hinders their discovery and elucidation of the conserved antigenic structure of highly mutable pathogens. Foot-and-mouth disease virus (FMDV) severely infects pigs and exists as seven distinct serotypes without cross-protection. In this study, we present a strategy for rapid discovery of cross-serotype bnAbs by constructing of specific B-cell repertoires and exploring of common clonotypes. We originally isolated porcine bnAbs against FMDV serotypes O, A and/or Asia1, and proved for the first time that receptor binding region in VP1 can induce the production of cross-serotype bnAbs. The evolution of cross-serotype bnAbs reveals how somatic hypermutations balance the breadth and potency of neutralization. FMDV bnAbs employ dual neutralization mechanisms to neutralize viruses via receptor mimicry and destabilizing the viral particles. This study provides valuable guidelines that facilitate the isolation of bnAbs against the highly variable viruses and aid in designing a universal FMDV vaccine.

## Introduction

Foot-and-mouth disease virus (FMDV) is a member of the *Picornaviridae* family, and a highly pathogenic virus that affects pigs, cattle, sheep and other cloven-hoofed animals. Seven immunologically distinct serotypes (O, A, Asia1, C, SAT1, SAT2, and SAT3) are described worldwide and clustered into distinct genetic lineages with approximately 30%-50% difference in the VP1 gene [1]. O, A, and Asia1 are the most widespread serotypes of FMDV reported to the World Organization for Animal Health (WOAH). Serotype O exhibits the highest prevalence and has been found in various regions including Africa, southern Asia, the Far East, and South America [1]. According to the nucleotide divergence threshold of 15%-20% in the VP1 gene and the epidemic region, there are eleven genetically and geographically distinct evolutionary lineages (topotypes) in serotype O [2]. Serotype A comprises three topotypes but demonstrates significant antigenic diversity [3,4]. The serotype Asia1 is endemic in parts of Asia [5]. There is no cross-protection among serotypes conferred by vaccination or previous infection [6]. The ability to readily mutate is an intrinsic characteristic of RNA viruses, enabling them to evade antibodies neutralization. This characteristic is noticeable in FMDV, which is one of the oldest viruses affecting livestock and has persisted for over a century. The continuous viral mutation and emergence of epidemic variants make the eradication of FMDV exceptionally challenging.

In the context of the genetic diversity of viral variations, immunologists endeavor to identify and characterize broadly neutralizing antibodies (bnAbs). However, these antibodies constitute a rare population within the body and often require repeated antigen stimulation and years of accumulation of somatic hypermutation (SHM) to evolve and develop, as exemplified by the timeframe of 2–4 years necessary for production of bnAbs against HIV [7,8]. Moreover, within the immune milieu, antibodies not only neutralize the pathogens but also select viruses with mutations that enable escape from antibody recognition, thereby hindering the development of highly broad-spectrum neutralizing antibodies [9–11]. Therefore, dissecting the process of bnAbs formation and resolving their recognized structures can provide crucial targets for designing broad-spectrum vaccines.

The bnAbs typically exert extensive virus-neutralizing activity by targeting vulnerable sites or conserved regions on viral particles [12,13]. Consistently, the highly conserved structure of

the virus antigen should confer the survival of the virus under evolution, rendering it a potential target for bnAbs against multiple variations [12,14].

Despite extensive studies utilizing monoclonal antibodies derived from mice, the discovery of bnAbs against FMDV from large animals remains elusive. So far, we have only isolated a bovine-derived bnAb with an ultralong heavy chain complementarity-determining region 3 (HCDR3) that is capable of neutralizing FMDV serotypes O and A [15]. Meanwhile, another research team reported the identification of a broad-spectrum single-domain antibody derived from alpaca [16]. These key findings from previous studies suggest that the identified bnAbs are peculiar and predominantly recognize antigens using a single structural domain. The potential for traditional bivalent antibodies to possess broad-spectrum neutralization capabilities against FMDV still remains unknown. Thus, high-throughput approaches are imperative in investigating the conserved antigenic structures among different serotypes and identifying bnAbs across various serotypes of FMDV.

In this study, we devised an approach for the precise extraction of bnAbs by constructing and dissecting of different antigen-specific B cell receptors (BCRs) repertoires derived from the same animal. The FMDV serotypes O and A specific class-switched B cells were respectively purified from PBMCs in the vaccinated pig and submitted to high-throughput transcript sequencing for production of BCR repertoires. As expected, from the shared BCR clonotypes across the two repertoires, the cross-serotype bnAbs against multiple epidemic FMDV strains were found. Evolutionary processes of porcine bnAbs suggested their B cells origins and how SHM balance neutralization breadth and potency. Moreover, the recognized FMDV antigenic structures by bnAbs reveal a direct and efficient approach for neutralizing antibodies that mimics receptor recognition and destabilizes viral particles, potentially representing a universal mechanism of bnAbs against highly mutable strains with cross-serotype neutralization capability.

## Results

### Production and characterization of FMDV-specific porcine B-cell repertoires

For induction of broad neutralization antibodies against FMDV, the pig was sequentially vaccinated with the inactivated 146S antigens of serotypes O and A strains, following the immunization schedule shown in Fig 1A. The serum samples of pigs at different time points after immunizations were tested using cross-neutralization assays against FMDV serotypes O, A and Asia1. The data showed that the first two immunizations with serotype O vaccine did not induce obvious cross-serotype neutralizing antibody response in pigs, showing less neutralization against serotype A (titer <1:16) and Asia1(titer <1:8) (Fig 1B). However, when boosted with mixture of serotype O/A vaccine, the pig obviously produced high titer bnAbs against Asia1, suggesting that the serotype Asia1 cross-neutralization antibody can be induced by sequential vaccination with serotypes O and A antigen in pig.

To selectively isolate neutralizing antibodies against FMDV from the diverse antibody secreting B cells present in the porcine immune system, we developed a protocol combining magnetic separation to deplete non-B cells and fluorescence-activated cell sorting (FACS) to sort antigen-specific B cells (Fig 1C). This approach effectively enriched a highly purified population of FMDV-specific B cells from peripheral blood mononuclear cells (PBMCs) isolated from vaccinated pig. As shown in Fig A in S1 Appendix, the FMDV-specific class-switched B cells were a scarce population and account for only about 0.02% in total porcine PBMCs. After depletion of T cells, NK cells, monocytes and IgM$^+$ B cells from PBMCs using magnetic separation, the proportion of specific B cells increased to 0.4%, up to 20-fold enrichment (Fig A,

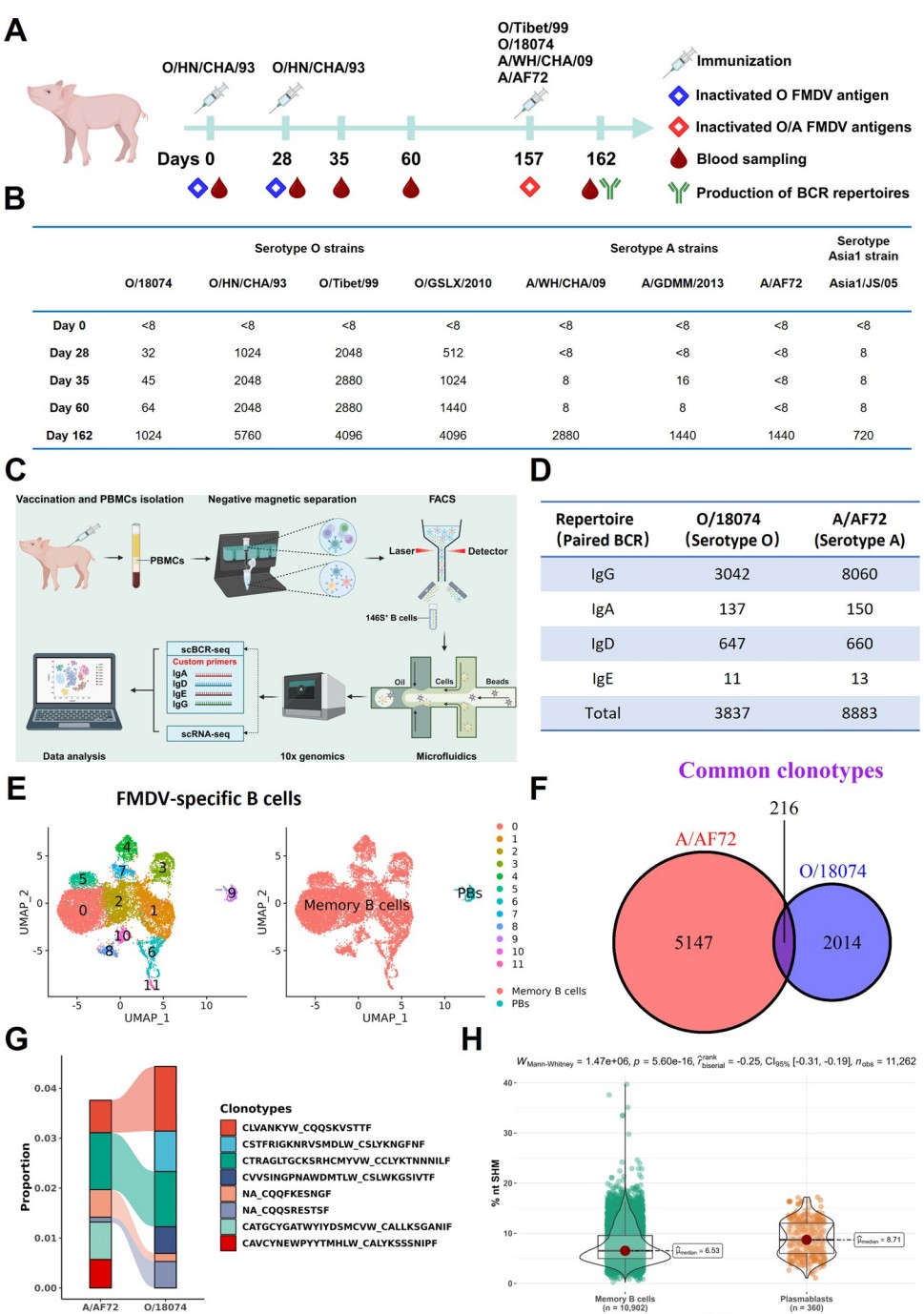

**Fig 1. Production and molecular characterization of FMDV-specific porcine B-cells repertoires.** **(A)** Time flow diagram for sequential vaccinations, blood sampling, and BCR repertoire production in the pig. Created in BioRender. Chen, Y. (2024) BioRender.com/a86r032. **(B)** The dynamic cross-neutralization antibodies titers against representative strains of serotypes O, A and Asia1 in sera sampled at different time after vaccination. **(C)** Workflow of production of FMDV-specific porcine BCR repertoire by purification of antigen-specific B cells via fluorescence-activated cell sorting (FACS) after magnetic depletion separation from PBMCs, and submission to single cell BCR and transcripts sequencing by 10×genomics using custom porcine BCR primers. Created in BioRender. Chen, Y. (2024) BioRender.com/f28f091. **(D)** Size and isotypes of porcine BCRs specific to FMDV serotypes O (O/18074 strain) and A (A/AF72 strain). **(E)** UMAP plot of unsupervised clustering of FMDV-specific porcine B cells, comprising of memory B cells and PBs (plasmablasts). **(F)** The numbers of common clonotypes shared in serotypes O and A specific BCR repertoires. **(G)** Proportion and connection of the top six high frequency of clonotypes respectively existing in serotypes O and A specific BCR repertoires. The presence of "NA" signifies the unavailability of CDRH3 in the B cell due to unsuccessful

sequencing of the heavy chain. **(H)** Comparing of SHM difference of porcine VH derived from memory B cells and plasmablasts. The statistical analysis was performed using non-parametric Mann-Whitney test in R program and showed the medians in violin plot. The value P<0.001 represents an extremely significant difference between two samples.

panel B in S1 Appendix). Subsequently, the pre-enriched B cells were loaded to sort the FMDV-specific B cells respectively using the O/18074 and A/AF72 146S as bait antigens by FACS, and serotype A- and O-specific B cells populations up to 90% purity were obtained (Fig A, panel C in S1 Appendix). Next, using 10×Genomics platform, the purified B cells were submitted to single cell V(D)J sequencing after PCR amplification of porcine BCR genes using species-specific primers designed in house (Table A in S1 Appendix). Finally, we successfully established the porcine BCR repertoires specific to FMDV serotypes O and A, respectively.

The porcine BCR repertoires contain 12720 paired variable regions of heavy and light chain (VH and VL) that consist of 3837 BCRs against O/18074 strain and 8883 BCRs against A/AF72 strain (Fig 1D). The proportion of each class-switched isotype showed the IgG antibody was dominated in pig against both FMDV serotypes O and A, accounting for 87% of total isotypes. The specific IgA antibody, which associated with the mucosal immunity, also appeared in circulating peripheral blood system, showing 2% proportion in these antibody repertoires (Fig 1D). Pairing analysis of BCR and transcriptome reveals the porcine antibodies were derived from plasmablasts and heterogenous memory B cells (Fig 1E and Supporting text, Section A in S1 Appendix).

To dissect the antibody diversity in pig, the clonotype was conventionally defined by a cluster of B cells with identical amino acids (AAs) of the third complement determination regions of paired antibody heavy and light chains (HCDR3 and LCDR3). There is equal proportion of expanded and singleton clonotypes in each of the serotype-specific repertoire, reflecting similar configuration of immunodominant epitopes between the two serotypes (Fig D, panel A-B in S1 Appendix). Notably, we found 216 common clonotypes shared in both O/18074-specific and A/AF72-specific repertoires, and these common clonotypes suggested the existence of the conserved antigen structure on different FMDV serotypes (Fig 1F). In addition, the top highest frequency clonotypes in both O and A repertoires also contained four of the clonotypes shared between O and A (Fig 1G). This observation appears to contradict the infrequency of broadly reactive responses, which could potentially be attributed to the sequential immunization program employed in this study.

Antibody affinity maturation is related to the increase of somatic hypermutation (SHM) that results in the production of antibodies with higher affinity or binding strength to pathogens[17]. Notably, plasmablasts exhibited significantly higher levels of total SHM compared to memory B cells, suggesting that antibody affinity maturation may occur during the development process from memory B cells to plasmablasts in porcine antibody response (Fig 1H). In detail, the memory B cells mutation frequencies cover a wider range than plasmablasts, and include some unmutated B cells as well as B cells with extremely high mutation frequencies (>20%) whereas the plasmablasts have no unmutated B cells and also no B cells with >20% mutation frequency (Fig 1H and 1E, panel A-B in S1 Appendix). We further checked the SHM accumulation in pig after FMDV vaccination and found total SHM in O/18074-specific repertoire was obviously higher than that in A/AF72-specific repertoire, and this was reflected concretely in the difference of isotypes IgG and IgA, not IgD (Fig E, panel C-F in S1 Appendix). Antibody class switch recombination (CSR) was also associated with the maturation of natural antibodies and improvement in SHM [18, 19]. In O/18074-specific repertoire, the SHM of class-switched isotypes IgG and IgA exhibited a significant increase to 10–11% compared to that of the IgD isotype (Fig E, panel G in S1 Appendix). However, the phenomenon was inconsistent in the A/AF72-specific repertoire that displayed almost the same SHM between IgG

and IgD isotypes. For the IgA isotype, both of repertoires have rather high SHM accumulation up to about 10%. The disparity in SHM between the two repertoires indirectly reflects variations in humoral response to serotypes O and A strains in the vaccinated pig, thereby providing information for isolating bnAbs against FMDV.

### The common clonotypes shared in porcine antibodies repertoires uncover broadly cross-serotypes neutralizing antibodies against Asia1 FMDV

From a pool of 216 common clonotypes shared in A/AF72-specific and O/18074-specific repertoires, we selected the top 10 clones with SHM and the top 10 high frequency clones to identify cross-serotype bnAbs against FMDV (Fig 1F). The 20 porcine-derived monoclonal antibodies (mAbs) were successfully expressed in 293F cells, and the purified mAbs were tested for FMDV reactivity by immunofluorescence assay (IFA) (Fig F in S1 Appendix) and enzyme-linked immunosorbent assay (ELISA) (Fig G in S1 Appendix). As expected, each of these mAbs exhibited reactivity towards FMDV serotypes O (O/18074 isolate) and A (A/AF72 isolate), with some variations in their binding capacities as assessed by IFA and/or ELISA. Although faint fluorescence signals were observed for mAbs pOA-3 and pOA-10 in IFA, their binding activities (<1 μg/ml) were confirmed through ELISA analysis. These data confirmed that the established porcine antibody repertoires have relatively high specificity.

Next, we verified the viral neutralization breadth and potency of these porcine-derived mAbs using virus neutralization assay (VNT) against the representative epidemic strains of FMDV serotypes O, A and Asia1 in China, including the O/HN/CHA/93 and O/18074 strains of Cathay topotype, O/Tibet/99 strain of ME-SA topotype, O/GSLX/2020 strain of SEA topotype, A/AF72 strain of A22 lineage, A/WH/CHA/09 and A/GDMM/2013 strains of SEA97 lineage as well as Asia1/JS/05 strain. As shown in Fig 2, most porcine-derived mAbs (15/20) exhibited virus neutralization activity against at least one representative strain. Of which, ten mAbs (pOA-1, pOA-2, pOA-6, pOA-7, pOA-8, pOA-9, pOA-13, pOA-17, pOA-19 and pOA-20) showed broadly neutralizing activity against six epidemic strains in both serotypes O and A, depicting the cross-serotype FMDV bnAbs feature. Notably, seven of these bnAbs exhibited potent neutralization activity against the Asia1 FMDV (Asia1/JS/05 strain), although the bnAbs derived pig did not receive the Asia1 antigen vaccination. This indicates that vaccinations with serotypes O and A antigen contributed to the formation of bnAbs.

Interestingly, although the A/AF72 strain was used as bait antigen to establish serotype A-specific antibody repertoire, all these bnAbs failed to neutralize the A/AF72 strain, indicating that the ancient A/AF72 in A22 lineage may exist big difference in antigen structure to currently epidemic strains. The VN titers of these bnAbs against different FMDV strains showed certain differences and ranged from <1 μg/ml to 50 μg/ml. The pOA-2 showed potently neutralizing activity against O/18074 strain (<1 μg/ml). Meanwhile the pOA-7 and pOA-8 showed potently neutralizing activity against O/Tibet/99 strain (<1 μg/ml). However, these porcine-derived bnAbs exhibited comparable average virus neutralization (VN) titers against FMDV serotypes O, A and Asia1, thereby offering valuable tools for investigating conserved antigen structures across different serotypes.

### Porcine cross-serotype bnAbs predominantly target the conserved "RGDL" motif on VP1 with the first residue after arginine-glycine-aspartic motif (RGD+1) being a functional determinant, potentially mimicking integrin receptors recognition

To elucidate the cross-serotype antigenic feature of FMDV, we initially conducted a western-blot (WB) assay using denatured FMDV 146S antigen to distinguish between linear or

| mAb | Serotype O | | | | Serotype A | | | Serotype Asia1 | Characteristics of porcine BCR | |
| --- | --- | --- | --- | --- | --- | --- | --- | --- | --- | --- |
| | O/18074 | O/HN/CHA/93 | O/Tibet/99 | O/GSLX/2010 | A/WH/CHA/09 | A/GDMM/2013 | A/AF72 | Asia1/JS/05 | #Frequency O(A) | SHM (% nt) |
| pOA-1 | 23.68 | 7.79 | 1.38 | 3.90 | 9.84 | 9.84 | >50 | >50 | 65(72) | 13.40 |
| pOA-2 | 0.82 | 11.17 | 1.76 | 14.10 | 45.26 | 39.09 | >50 | >50 | 55(156) | 11.78 |
| pOA-3 | 36.43 | >50 | 40.00 | >50 | >50 | >50 | >50 | >50 | 19(13) | 5.25 |
| pOA-4 | 45.56 | 37.27 | 1.07 | 10.25 | 29.29 | >50 | >50 | 44.44 | 17(15) | 13.08 |
| pOA-5 | >50 | >50 | >50 | >50 | >50 | >50 | >50 | >50 | 16(15) | 8.90 |
| pOA-6 | 6.38 | 15.79 | 4.69 | 13.64 | 2.46 | 3.90 | >50 | 1.90 | 13(8) | 9.39 |
| pOA-7 | 8.59 | 7.40 | 0.79 | 5.23 | 3.79 | 12.67 | >50 | 7.29 | 12(13) | 13.71 |
| pOA-8 | 3.03 | 4.29 | 0.95 | 4.89 | 1.73 | 7.67 | >50 | 5.58 | 12(10) | 13.08 |
| pOA-9 | 4.15 | 7.27 | 1.82 | 4.38 | 2.77 | 5.00 | >50 | 17.34 | 12(7) | 11.21 |
| pOA-10 | >50 | >50 | 23.53 | 18.18 | 18.18 | 18.18 | >50 | 9.69 | 12(8) | 9.58 |
| pOA-11 | 14.13 | 9.30 | 2.07 | 4.14 | >50 | 44.75 | >50 | >50 | 10(14) | $20.19 |
| pOA-12 | >50 | >50 | >50 | >50 | >50 | >50 | >50 | >50 | 3(2) | 18.46 |
| pOA-13 | 1.87 | 10.27 | 5.26 | 5.73 | 4.56 | 7.20 | >50 | 6.68 | 1(1) | 16.67 |
| pOA-14 | >50 | >50 | >50 | >50 | >50 | >50 | >50 | >50 | 1(3) | 16.57 |
| pOA-15 | 21.67 | >50 | >50 | >50 | >50 | >50 | >50 | >50 | 11(5) | 15.77 |
| pOA-16 | >50 | >50 | >50 | >50 | >50 | >50 | >50 | >50 | 2(3) | 15.63 |
| pOA-17 | 7.06 | 6.73 | 17.27 | 17.27 | 11.88 | 23.75 | >50 | 29.69 | 1(1) | 15.03 |
| pOA-18 | >50 | >50 | >50 | >50 | >50 | >50 | >50 | >50 | 1(1) | 1631 |
| pOA-19 | 7.29 | 7.69 | 1.92 | 3.29 | 11.61 | 26.91 | >50 | >50 | 24(2) | $39.55 |
| pOA-20 | 3.12 | 11.91 | 5.19 | 7.32 | 5.91 | 8.87 | >50 | 7.07 | 5(19) | 15.76 |

**VN titers (µg/ml)**
- <1.00
- 1.00–5.00
- 5.00–10.00
- 10.00–25.00
- 25.00–50.00
- >50

**Fig 2. Neutralization titer and breadth of porcine monoclonal antibodies with common clonotypes shared in both O/18074-specific and A/AF72-specific repertoires.** Porcine mAbs against representative strains of FMDV serotypes O, A and Asia1 tested by microneutralization assay. Values are virus neutralization titer (VN) in µg/ml and shown with color bar on the right. Frequency indicated the frequency of clonotypes for each mAb that appeared in O/18074-specific and A/AF72-specific (in parenthesis) repertoires. SHM indicated frequency of replacement and silent mutations in full variable region sequence of heavy chain (VH), which was calculated by SHazaM using the program Immcantation. $ noted the SHM data of this mAb was calculated basis on V gene segment of heavy chain.

conformational antigen epitopes recognized by these 10 porcine-derived bnAbs. For both serotypes O and A, the majority of bnAbs (9/10) recognized the denatured 146S antigen and bound to VP1. Only one bnAb pOA-2 (1/10) did not bind to denatured viral antigens, indicating recognition of a conformational antigen epitope on FMDV (Fig 3A and 3B).

Furthermore, a series of GST-VP1 truncated proteins were respectively probed using the nine bnAbs. WB results showed these bnAbs bound to all the truncated proteins within

G-H loop, not the C-terminus (Fig 3B). To determine the minimal requirement for the nine bnAbs recognition, the VP1 G-H loop ([143]NVRGDLQVLAK[154]) peptide was further truncated from the C-terminus; as demonstrated by WB, the shorter peptide, [143]NVRGDL[148], showed reactivity with the nine bnAbs, whereas the further truncated peptide, [143]NVRGD[147], showed no binding ability with these bnAbs (Fig 3C). Subsequently, [143]NVRGDLQVLAK[154] was further truncated from its N-terminus and respectively subjected to WB (Fig 3C). Thus, the peptide, [145]RGDL[148], was determined to be the minimal unit for recognition by nine bnAbs (Fig 3D). Subsequently, each residue of the peptide, [143]NVRGDLQVLAK[154], as a GST fusion protein, was singly substituted with alanine, and WB showed that substitution at position 143, 145–148 and 151 reduced the binding of these bnAbs (Fig 3E). Particularly, single substitution occurring at position 146 and 147 completely abolished the binding activity of all these bnAbs. Additionally, alignment of all these representative strains in serotypes O, A and Asia1 shows that the common residues (RGDL) are strictly conserved (Fig 3F). Further analysis of available FMDV VP1 sequences also revealed the RGDL were extremely constant with conservation of 99.6% and 87.2% in serotypes O and A (Fig H in S1 Appendix). Altogether, the residues on VP1 at positions 143, 145–148 and 151 form the epitopes and the core residues "RGDL" on the protruding region of G-H loop are key determinants for these cross-serotype bnAbs recognition, revealing a cross-serotype antigen site that exists on FMDV (Fig 3G).

FMDV employs integrin (generally $\alpha v\beta 6$) as primary receptor to entry epithelial cells, causing infection in susceptible host [20–22]. The conserved arginine-glycine-aspartic (RGD) motif in the exposed G-H loop of VP1 facilitated the virus to bind the integrin receptor[23]. The flexibility of G-H loop of VP1 on viral surface made the conformation of virus-receptor complex diversiform and four different conformations have been identified by the cryo-EM complex structures of FMDV-$\alpha v\beta 6$ [24]. We reanalyzed the four poses of FMDV-$\alpha v\beta 6$ resolved by Kotecha et al. to reveal interface residues that involved in formation of hydrogen bonds and salt bridges. Coincidentally, the interaction residues in these conformations mostly overlapped with the FMDV cross-serotype epitopes, including the pose A (residues143N, 145R and 147D on VP1) and pose A' (residues at position 143N, 145R and 148L on VP1) of Panasia plus integrin, as well as the pose A (residues 145R, 147D and 151L on VP1) and pose A' (residues 147D and 151L on VP1) of O1M plus integrin (Fig 3H and Table B in S1 Appendix). The similar binding mode between FMDV-receptor and FMDV-bnAb suggests these bnAbs may employ the strategy of mimicking receptor recognition to potentially neutralize against all the diverse strains in multiple serotypes via blocking viral attachment.

To identify crucial functional determinants involving these epitopes on viable viral particle, the neutralization escape mutants were selected for the 10 bnAbs against O/HN/CHA/93. Most of bnAbs (9/10) selected for the variations at the RGD+1 or +2 positions on VP1, whereas the other one bnAb (pOA-2) made the variation at 68 position on VP2 (Supporting text, Section B in S1 Appendix). Next, the single mutants at position VP2 68, RGD+1 and RGD+2 were individually constructed based on the whole structure proteins of O/HN/CHA/93 and A/WH/CHA/09 strains separately. Cross-neutralization assay revealed that the RGD+1 residue (L) was a key conserved antigenic determinant on both serotypes O and A and represented a novel cross-serotype antigen site on FMDV (Fig 3I and 3J). Meanwhile, RGD+2 residue could be the serotype-specific antigenic determinant and biasedly targeted by the intraserotype bnAbs [25–28]. Additionally, the rescued FMDV VP2 68 (D→N) single mutations in both serotypes O and A completely escaped the neutralization by only bnAb pOA-2 and contrastively the RGD+1/+2 mutations did not affect the neutralization by pOA-2, consolidating the VP2 68 (D) is a key determinant that represents another one functionally independent cross-serotype antigen site on FMDV (Fig 3I and 3J).

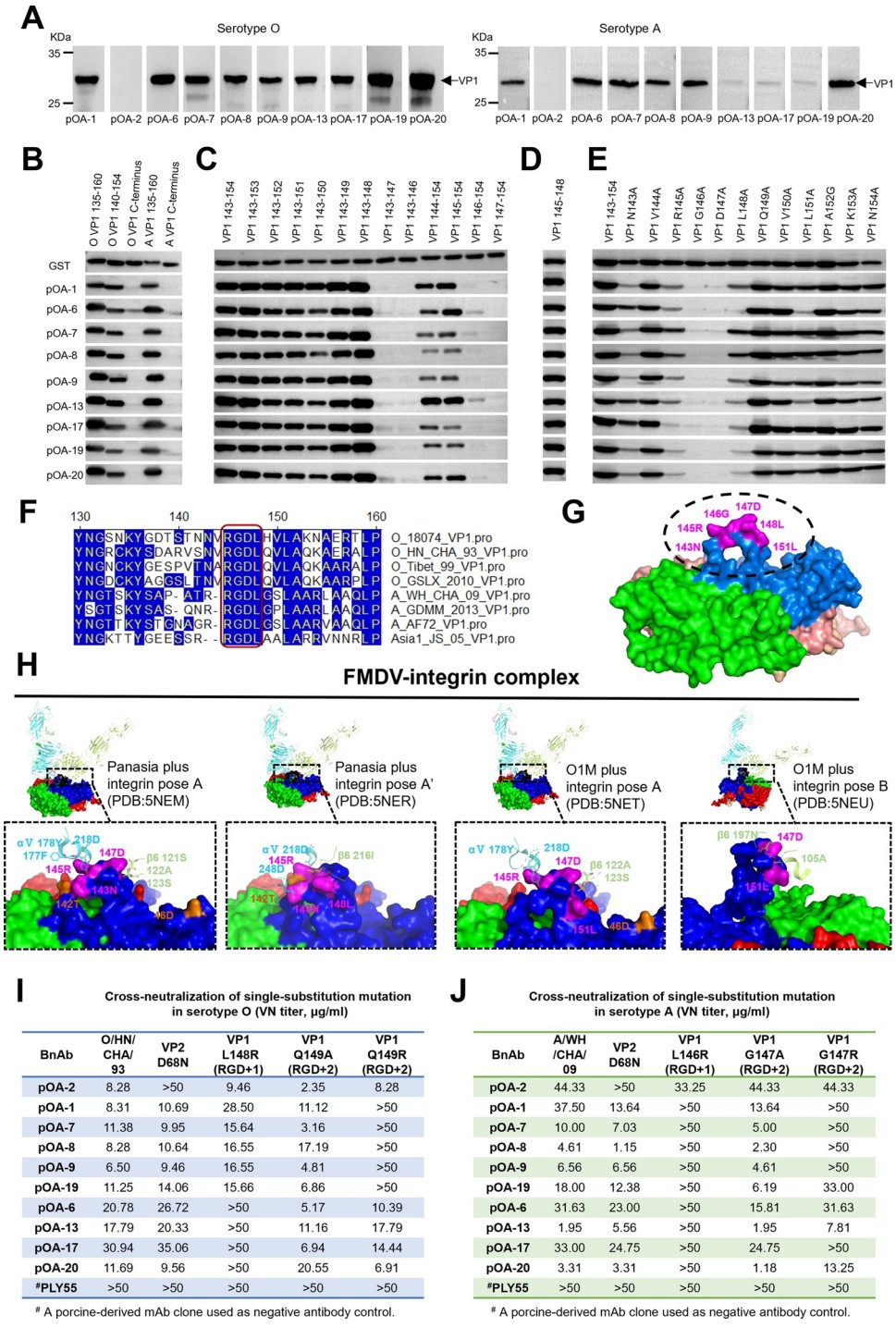

**Fig 3. Identification of porcine bnAbs binding epitopes and critical determinants as well as their overlap with integrin receptor (αvβ6) recognizing site.** (A) Western blot identifying the reactivity of ten bnAbs with the denatured 146S antigens of FMDV serotypes O and A.(B-E) WB identification of linear epitope and critical residues affecting the binding with porcine bnAbs using GST-fused truncated VP1 of FMDV (O/18074 strain). The GST-fused G-H loop and C-terminus of VP1 (B), truncated epitopes (C), minimum epitope (D) and single alanine substituted epitopes (E) were respectively probed with the nine bnAbs individually. The HRP-labeled anti-GST mAb was used as control for assurance of loading equal amounts of antigen. (F) Sequence alignment of VP1 G-H loop of representative strains of FMDV serotypes O, A and Asia1. (G) Mapping of epitopes on G-H loop of FMDV identified by the nine cross-serotype bnAbs. (H) The overlapped residues between antigenic determinants of cross-serotype bnAbs and interface residues of FMDV-receptor (αVβ6) complexes were displayed with pink on surface of FMDV protomer. Other

interface residues were colored in orange. The αV and β6 chains of integrin receptor were respectively colored in cyan and limon. The VP1, VP2, VP3 and VP4 forming one protomer were colored in purple, green, red and yellow, respectively. The FMDV-receptor complexes were resolved by Kotecha A et al. in previous study[24]. **(I,J)** Identification of the recognizing key antigenic determinants of ten cross-serotype bnAbs using the rescued single mutation FMDV strains.

## The cryo-EM complexes confirm a novel cross-serotype antigen structure at the three-fold axis on FMDV, involving the key determinant on VP2 68 (D) in both serotypes A and O

Next, we focused on the cross-serotype antigen structure and determined the structures of pOA-2 single-chain fragment variable (scFv) in complex with O/18074 (FMDV-O18-POA2) and A/WH/CHA/09 (FMDV-AWH-POA2), respectively. The cryo-EM micrographs indicated that scFv attached to the locations around the three-fold axis on FMDV capsids, and total 60 copies scFv were bound to each particle (Fig 4A and 4B). The final resolution of the cryo-EM reconstruction was estimated by the FSC 0.143 cutoff to be 2.44 Å for the FMDV-O18-POA2 complex and 2.52 Å for the FMDV-AWH-POA2 complex (Fig I in S1 Appendix). In both cases, the cryo-EM densities were of sufficient quality to allow for atomic modeling of the FMDV capsid proteins and the variable loops of the scFv antibodies that are responsible for virus recognition.

The pOA-2 interacts with FMDV-O18 VP2 and VP3 in two protomers by its heavy and light chains together (VH and VL), creating a buried surface area of 1008.1 $Å^2$ on the three-fold axis, as calculated by PISA (Fig 5A and Table D in S1 Appendix). The antibody VH and VL mediate equal contact area with complex FMDV-O18-POA2, and make 16 hydrogen bonds and 5 salt bridges at the interface (Table F in S1 Appendix).

On one protomer, residues in VP2 βB ($_{O18-VP2}$H65, $_{O18-VP2}$L66, $_{O18-VP2}$F67, $_{O18-VP2}$D68 and $_{O18-VP2}$G70), B-C loop ($_{O18-VP2}$T71, $_{O18-VP2}$N72 and $_{O18-VP2}$F75), βC ($_{O18-VP2}$R77), H-I loop ($_{O18-VP2}$P195 and $_{O18-VP2}$Q196) and βI ($_{O18-VP2}$K198) interact with residues in the six CDRs of pOA-2, ordinal HCDR3 ($_{VH}$K105, $_{VH}$R107 and $_{VH}$G103), LCDR2 ($_{VL}$Y51), LCDR1 ($_{VL}$Y33), LCDR3 ($_{VL}$Y92 and $_{VL}$N95), HCDR1 ($_{VH}$E33) and HCDR2 ($_{VH}$Q53) (Fig 5B to 5D). Meanwhile, residues in the VP3 B-C loop ($_{O18-VP3}$T68, $_{O18-VP3}$D69 and $_{O18-VP3}$S70) and H-I loop ($_{O18-VP3}$D195) from another protomer interact with residues in LCDR2 ($_{VL}$N53), FR3 ($_{VL}$S54 and $_{VL}$R55) and FR2 ($_{VL}$Y50) from light chain as well as the HCDR3 ($_{VH}$T102) (Fig 5E). Notably, the binding region of pOA-2 on VP2 and VP3 covers two different protomers that cross the interface between neighboring pentamers on FMDV (Fig 5F). The side chains of $_{O18-VP2}$H65 and $_{O18-VP2}$D68 form hydrogen bond contacts with the $_{VH}$K105 side chain. Additionally, the $_{O18-VP2}$D68 side chain also forms hydrogen bond contacts with the $_{VL}$Y51 side chain (Table F in S1 Appendix). To further validate the crucial determinants of FMDV serotype O for pOA-2, we substituted alanine for specific capsid residues involved in forming hydrogen bonds or salt bridges at the interface of the virus-antibody complex. A total of 8 single-substitution mutants were successfully rescued (Fig J, panel A in S1 Appendix) and assessed for neutralization potency with pOA-2. As shown in Fig 5G, mutations at position 68 on VP2 as well as positions 70 and 195 on VP3 significantly reduced antibody neutralization, indicating the three residues represented the key determinants on serotype O FMDV recognized by pOA-2.

The interaction of pOA-2 with FMDV-AWH is homologous to that observed in the FMDV-O18-POA2 complex and creates a buried surface area of 896.2 $Å^2$ on viral surface, mediating 55% and 45% contacts by antibody VH and VL, respectively (Table E in S1 Appendix). pOA-2 makes a total of 9 hydrogen bonds and 3 salt bridges contacts with

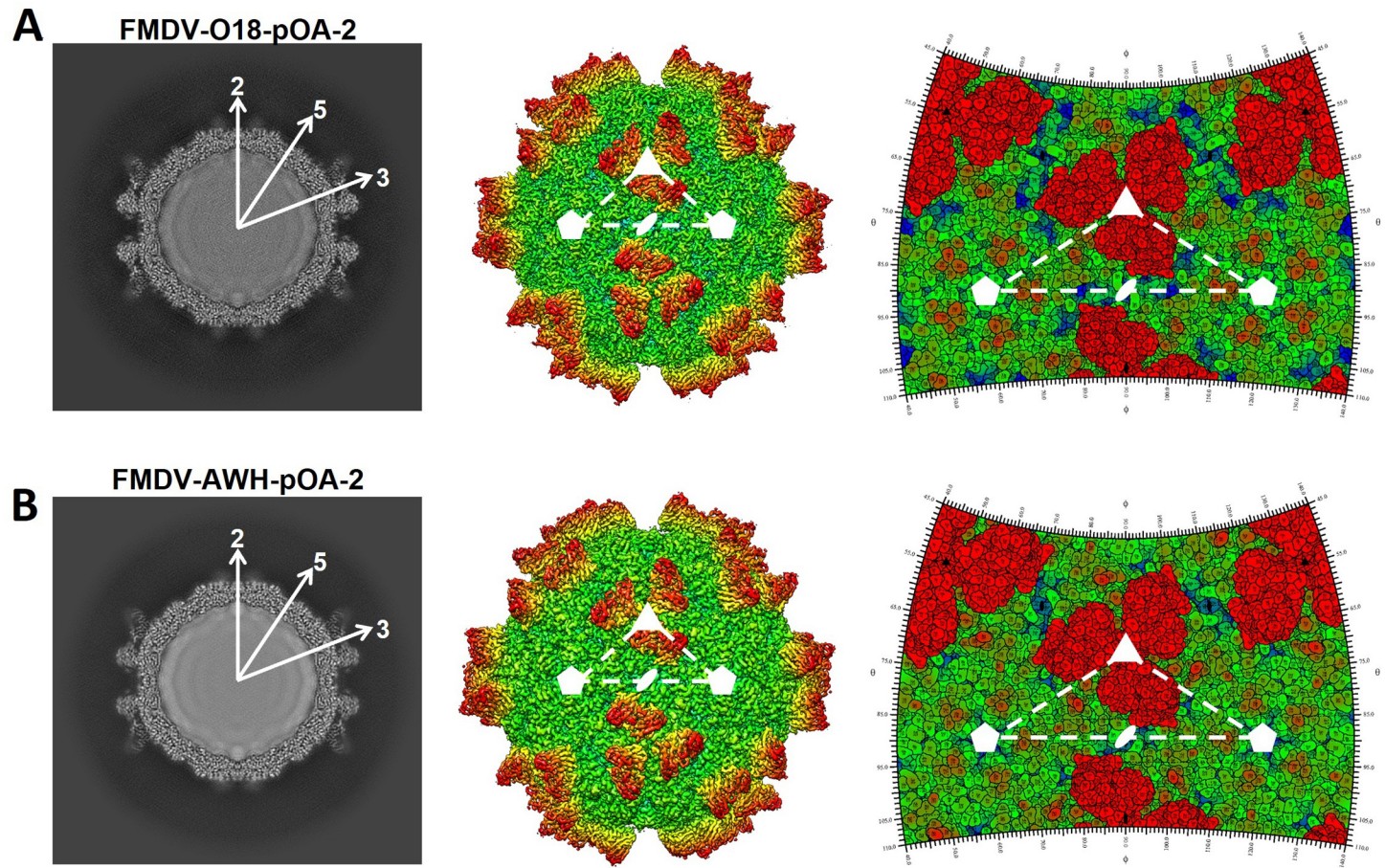

**Fig 4. Cryo-EM analysis of FMDV-O18-POA2 complex and FMDV-AWH-POA2 complex.** (A, B) The central cross sections, rendered images and footprints of FMDV-O18-POA2 complex (A) or FMDV-AWH-POA2 complex (B) are shown. The central cross sections obtained through cryo-EM maps are shown with icosahedral 2-, 3-, and 5-fold axes. In the rendered images, depth cueing with color is used to indicate the radius (< 120 Å, cyan; 140–160 Å, from green to yellow; > 180 Å, red). The icosahedral 5- and 3-fold axes are represented by pentagons and triangles, respectively. Footprints of pOA-2 on the FMDV surface were produced using RIVEM [67].

FMDV-AWH VP2 while only one hydrogen bond and one salt bridge contact with VP3, showing a biased interaction distribution on two adjacent protomers (Fig 6A and Table G in S1 Appendix). On one protomer, residues in VP2 βB ($_{AWH-VP2}$H65 and $_{AWH-VP2}$D68), B-C loop ($_{AWH-VP2}$T71 and $_{AWH-VP2}$D72), αB ($_{AWH-VP2}$K137), and H-I loop ($_{AWH-VP2}$S195) interact with residues in the five CDRs of pOA-2, ordinal HCDR3 ($_{VH}$K105), LCDR2 ($_{VL}$Y51), LCDR1 ($_{VL}$Y33 and $_{VL}$T31), LCDR3 ($_{VL}$N95) and HCDR2 ($_{VH}$D56) (Fig 6B to 6D). Meanwhile, residues in the VP3 B-C loop ($_{AWH-VP3}$D69 and $_{AWH-VP3}$E70) from another protomer interact with residues in FR3 ($_{VL}$R55 and $_{VL}$S54) and FR2 ($_{VL}$Y50) from antibody light chain (Fig 6E). Consistently, the VP2 68 (D) residue is also positioned within the central region of the virus-antibody interface. The side chains of $_{AWH-VP2}$H65 and $_{AWH-VP2}$D68 form hydrogen bond contacts with the $_{VH}$K105 side chain. Additionally, the $_{AWH-VP2}$D68 side chain also forms hydrogen bond contacts with the $_{VL}$Y51 side chain. (Table G in S1 Appendix). Furthermore, mutations at positions 65 and 68 on VP2 significantly reduced antibody neutralization. In particular, the mutations $_{AWH-VP2}$D68A and $_{O18-VP2}$D68A enabled complete escape of antibody-mediated neutralization (VN titer > 250 µg/ml) (Figs 6G and 5G, and J, panel B in S1 Appendix). Meanwhile, the sequence alignment shows that the common residue VP2 68 (D) that contacts pOA-2 is strictly conserved in all five

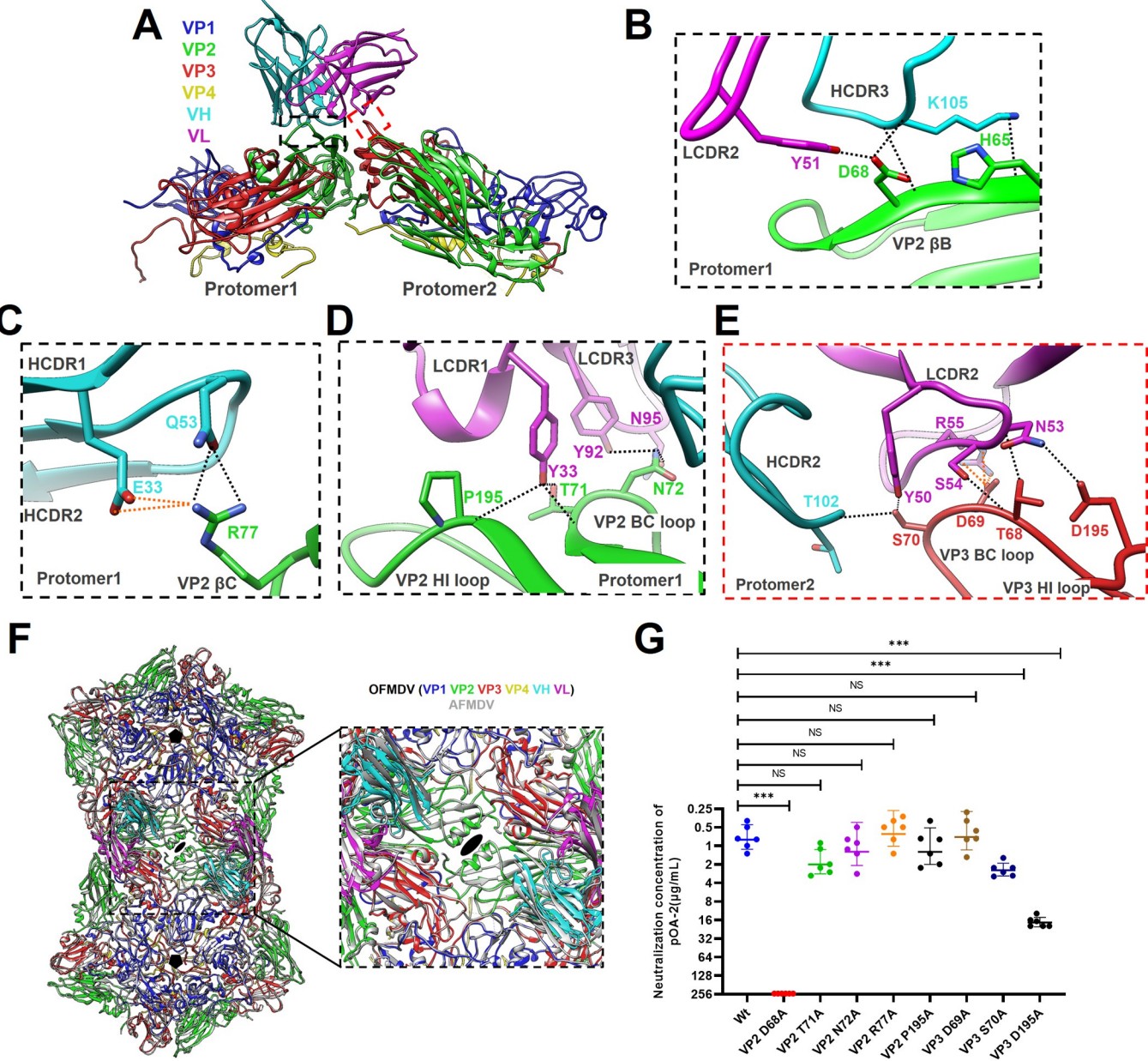

**Fig 5. Structure of the FMDV-O18-POA2 complex and key determinants on VP2 as well as VP3 of FMDV serotype O.** (A) Cartoon representation of two protomers showing the interaction interface between pOA-2 scFv and the capsid of the O/18074 strain. The heavy chain and light chain of pOA-2 are colored cyan and magenta, respectively. The capsid proteins VP1 to VP4 are colored blue, green, red and yellow, respectively. **(B-E)** Expanded views of the interaction interface highlighting the βB **(B)**, βC **(C)**, B-C loop and H-I loop **(D)** of VP2 within protomer 1 as well as the H-I loop **(E)** of VP3 within protomer 2. Presumable hydrogen bonds and salt bridges in the interaction interface are marked separately by black and orange dashed lines. **(F)** Binding regions of the bnAb pOA-2 that covers two different protomers that cross the interface between neighboring pentamers on FMDV. **(G)** The neutralizing efficacy of pOA-2 against the wild-type (O/18074) and its mutants were evaluated using a microneutralization assay. The neutralization concentration represented the lowest antibody concentration required to fully prevent CPE. The experiments were independently conducted in triplicate. The neutralization differences between wild-type (O/18074) and its mutants were determined using unpaired T test (Holm-Sidak method, with α = 0.05) in GraphPad Prism 7.2. *** Indicates an extremely significant difference to wild-type at P<0.001. NS indicates no significant difference.

representative strains of FMDV serotypes O and A ([Fig 6F]). The conservation of the residue appears more than 99% in available FMDV serotypes O and A sequences ([Fig 6H] and Table H in [S1 Appendix]). These findings demonstrate the conserved antigenic structure

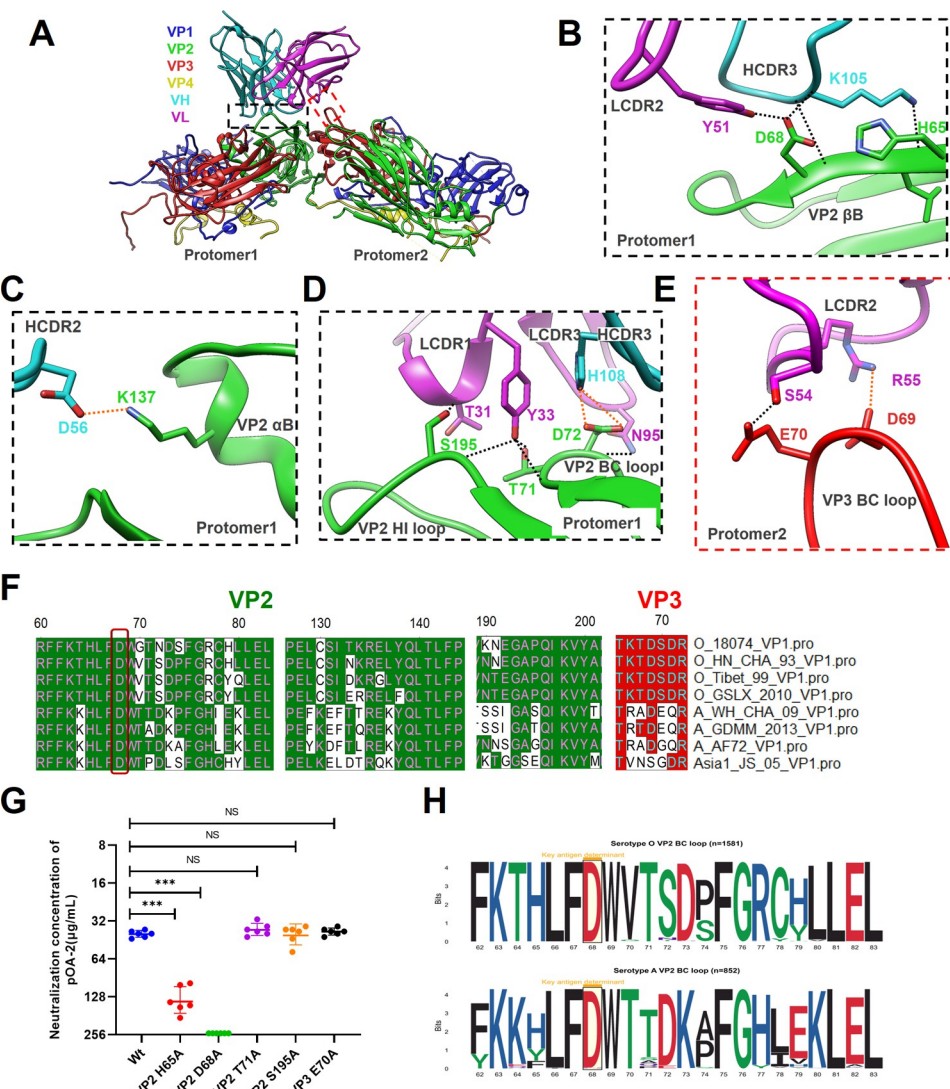

**Fig 6. Structure of the FMDV-AWH-POA2 complex and key determinants on VP2 as well as VP3 of FMDV serotype A. (A)** Cartoon representation of two protomers showing the interaction interface between pOA-2 scFv and the capsid of the A/CHA/WH/09 strain. The heavy chain and light chain of pOA-2 are colored cyan and magenta, respectively. The capsid proteins VP1 to VP4 are colored blue, green, red and yellow, respectively. **(B-E)** Expanded views of the interaction interface highlighting the βB **(B)**, αB **(C)**, B-C loop and H-I loop **(D)** of VP2 within protomer 1 as well as the H-I loop **(E)** of VP3 within protomer 2. Presumable hydrogen bonds and salt bridges in the interaction interface are marked separately by black and orange dashed lines. **(F)** Sequence alignment of VP2/VP3 of representative FMDV strains of serotypes O, A and Asia 1. **(G)** Neutralizing efficacy of pOA-2 against the wild-type (A/WH/CHA/09 strain) and its mutants were evaluated using a microneutralization assay. The neutralization concentration represented the lowest antibody concentration required to fully prevent CPE. The experiments were independently conducted in triplicate. The neutralization differences between wild-type (A/WH/CHA/09) and its mutants were determined using unpaired T test (Holm-Sidak method, with $\alpha = 0.05$) in GraphPad Prism 7.2. *** Indicates an extremely significant difference to wild-type at $P<0.001$. NS indicates no significant difference. **(H)** The sequence logo of VP2 B-C loop of FMDV serotypes O (numbers of full VP2 sequences = 1581) and A (numbers of full VP2 sequences = 852) available downloaded from national center for biotechnology information (NCBI) as of June 30, 2023.

surrounding the three-fold axis in both FMDV serotypes O and A, with VP2 68 (D) serving as a crucial antigenic determinant.

## The clonal evolution of porcine cross-serotype bnAbs-producing B cells lineages

Following antigen stimulation, naive B cells undergo proliferation and generate a clonally related lineage consisting of diverse antibody sequences characterized by the accumulation of somatic mutations [29]. To investigate the developmental pathways of FMDV bnAbs in vaccinated pig, we reconstructed evolutionary trees of 216 common BCR clonotypes using IgPhyML by immunogenetic annotation program Immcantation [30,31]. As shown in Fig 7A, these bnAbs were derived from five clonal lineages out of the total 120 clonal lineages (S1 Table) and employed four different heavy genes (IGHV1S5*01, IGHV1-4*01, IGHV1-10*01 and IGHV1S2*01) with HCDR3 lengths ranging from six to 16 amino acids (AAs). The notable observation is that out of the 9 cross-serotype bnAbs targeting the G-H loop, a total of eight antibodies employed the light chain gene IGKV2-10, indicating a convergence in their recognition of the VP1 G-H loop epitope. According to the evolutionary trees of the bnAbs lineages (Clone_105, Clone_76 and Clone_54), all seven bnAbs targeting the VP1 G-H loop are derived from terminally differentiated B cells that undergo multiple amplifications and accumulation a significant number of SHMs to acquire broad neutralization capabilities (Fig 7B).

## Somatic hypermutations balance the breadth and potency of neutralization

The factors governing the breadth and potency of neutralization in humoral immune response remain elusive. To address this, we dissected the evolutionary tree of bnAb pOA-2 containing clonal lineage (Clone_66) and evaluated the neutralizing efficacy of associated members including the unmutated common ancestor (UCA). As depicted in Fig 8A, the clonal lineage comprised six B cell members, and pOA-2 was positioned as an intermediate stage B cell origin. All tested five B cell members as well as their UCA all can bind to both FMDV serotypes O and A strains; although binding ability of UCA was slight weaker than those of the B cell members with accumulation of SHM (Fig 8B and 8C). However, the binding activity of UCA with FMDV antigens suggested the existence of precursor B cell for FMDV bnAbs development. Accumulation of different level SHMs were observed in all B cell members (Fig 8D). Initially, the UCA underwent extensive mutations up to 19 AAs across all complementarity-determining regions (CDRs) and framework regions (FRs) of the antibody heavy chain, resulting in the development of an early member 163 exhibiting broad neutralizing activity against four representative strains in serotype O. One additional substitution (102: L->V) in CDR3 of the member 163 formed a branchpoint (BP2) with two descendants. Subsequently, a further mutation at position 31 (S->N) in CDR1 on the BP2 led to the evolution of pOA-2, which significantly expanded its breadth of neutralization to encompass six representative strains in both serotypes O and A. However, subsequent one substitution (60: A->S) and additional more mutations on pOA-2 did not yield any additional increase in neutralization breadth for evolved members 162 and 165, inversely rendering them incapable of neutralizing A/WH/CHA/09, O/HN/CHA/93 or O/Tibet/99 strains. The cross-neutralization potency of members pOA-2, 162, and 165 was compared, revealing a significant enhancement in neutralization potency against the A/GDMM/2013 strain for members 162 and 165. Member 162 exhibited an approximate ten-fold increase (VN titer: 39.09 vs 3.88 μg/ml), while member 165 showed an approximate ninety-fold increase (VN titer: 39.09 vs 0.46 μg/ml) (Fig 8A and 8E). Collectively, the clonal evolution of this lineage suggests that SHM plays a crucial role in balancing the breadth and potency of neutralization against FMDV in porcine immune response; however, excessive SHMs can deviate from the trajectory of broad neutralization (Fig 8F).

**A**

### The immunogenetics annotation of bnAbs

| Sequence_id | Clone_id | H_V_gene | H_D_gene | H_J_gene | L_V_gene | L_J_gene | Length of HCDR3 (aa) | Length of LCDR3 (aa) | H_ Mutation (% nt) | L_ Mutation (% nt) |
|---|---|---|---|---|---|---|---|---|---|---|
| pOA-1 | 105 | IGHV1S5*01 | IGHD1*01 | IGHJ5*01 | IGKV2-10*02 | IGKJ2*02 | 6 | 8 | 13.40 | 13.23 |
| pOA-7 | 105 | IGHV1S5*01 | IGHD1*01 | IGHJ5*01 | IGKV2-10*02 | IGKJ2*02 | 6 | 8 | 13.71 | 14.46 |
| pOA-8 | 105 | IGHV1S5*01 | IGHD1*01 | IGHJ5*01 | IGKV2-10*02 | IGKJ2*02 | 6 | 8 | 13.08 | 12.00 |
| pOA-9 | 105 | IGHV1S5*01 | IGHD1*01 | IGHJ5*01 | IGKV2-10*02 | IGKJ2*02 | 6 | 8 | 11.21 | 12.00 |
| pOA-19 | NA | IGHV1S5*01 | NA | IGHJ5*01 | IGKV2-10*02 | IGKJ2*02 | 6 | 8 | NA | 13.85 |
| pOA-6 | 54 | IGHV1-4*01 | NA | IGHJ5*01 | IGKV2-10*02 | IGKJ2*01 | 7 | 9 | 9.39 | 6.25 |
| pOA-20 | 54 | IGHV1-4*01 | NA | IGHJ5*01 | IGKV2-10*02 | IGKJ2*01 | 7 | 9 | 15.76 | 7.74 |
| pOA-17 | 76 | IGHV1-10*01 | IGHD1*01 | IGHJ5*01 | IGKV2-10*02 | IGKJ2*02 | 11 | 9 | 15.03 | 8.63 |
| pOA-13 | 32 | IGHV1S5*01 | IGHD2*01 | IGHJ5*01 | IGLV8-16*01 | IGLJ2*01 | 9 | 11 | 16.67 | 11.90 |
| pOA-2 | 66 | IGHV1S2*01 | IGHD2*01 | IGHJ5*01 | IGLV8-19*02 | IGLJ2*01 | 16 | 10 | 11.78 | 12.69 |

**B**

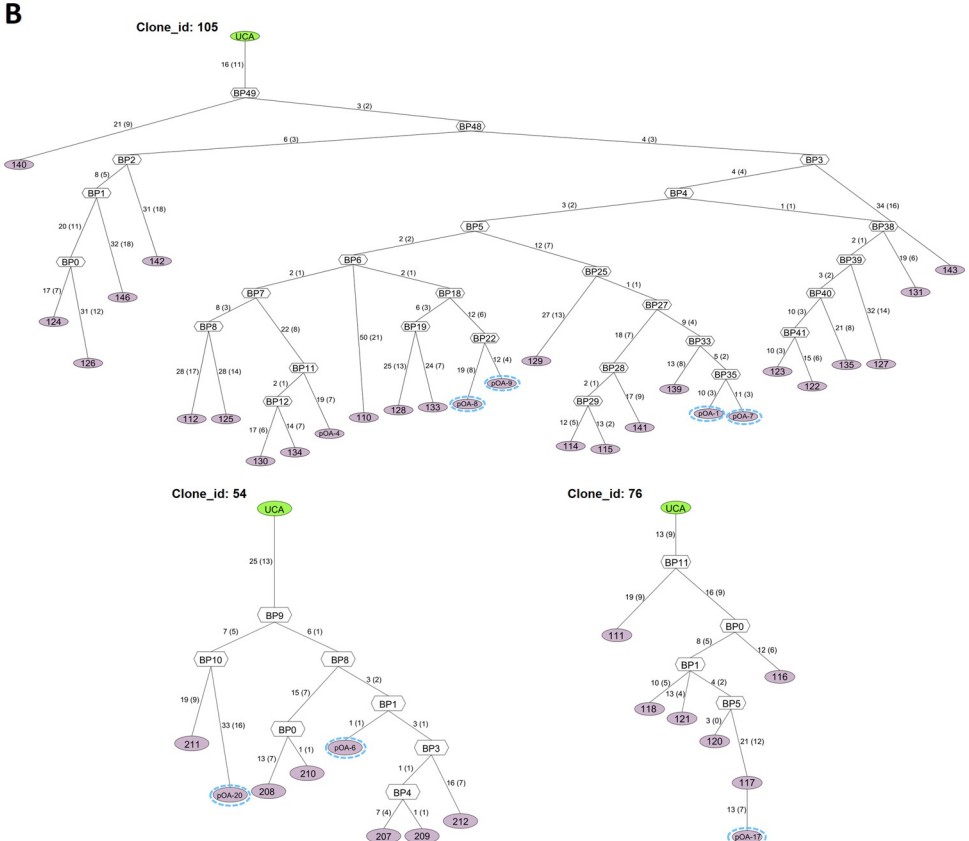

**Fig 7. The immunogenetics annotation of porcine bnAbs and the evolution of bnAbs-producing B cells lineages.**
**(A)** The immunogenetics annotation of porcine bnAbs using the program Immcantation. The "NA" indicated the data was not found in output of the program because of low match with porcine germline Ig gene segments in the IMGT reference directory. Basis on the difference in the V gene segment usage of light chain, 10 bnAbs were colored with blue, light blue, and pink backgrounds, respectively. Specifically, the 8 bnAbs employed the same light chain V gene IGKV2-10*02 on a blue background, while bnAb pOA-13 utilized IGLV8-16*01 on a light blue background. Additionally, bnAb pOA-2 employed IGLV8-19*02 on a pink background. **(B)** Phylogenetic trees of bnAbs-containing clonal lineages (Clones 105, 54 and 76) that were constructed by IgPhyML algorithm using VH sequences of the common clonotypes between serotypes O and A specific BCR repertoires and then visualized in AncesTree. The pink and green ovals respectively represent B cells and their related unmutated common ancestor (UCA). The number of nucleotide and amino acid mutations were written on the edge between each node/sequence (with amino acid mutations shown in parenthesis). Branch points (BPs) indicate the theoretical intermediate reconstructed sequences. The bnAb is circled with dotted line in each phylogenetic tree.

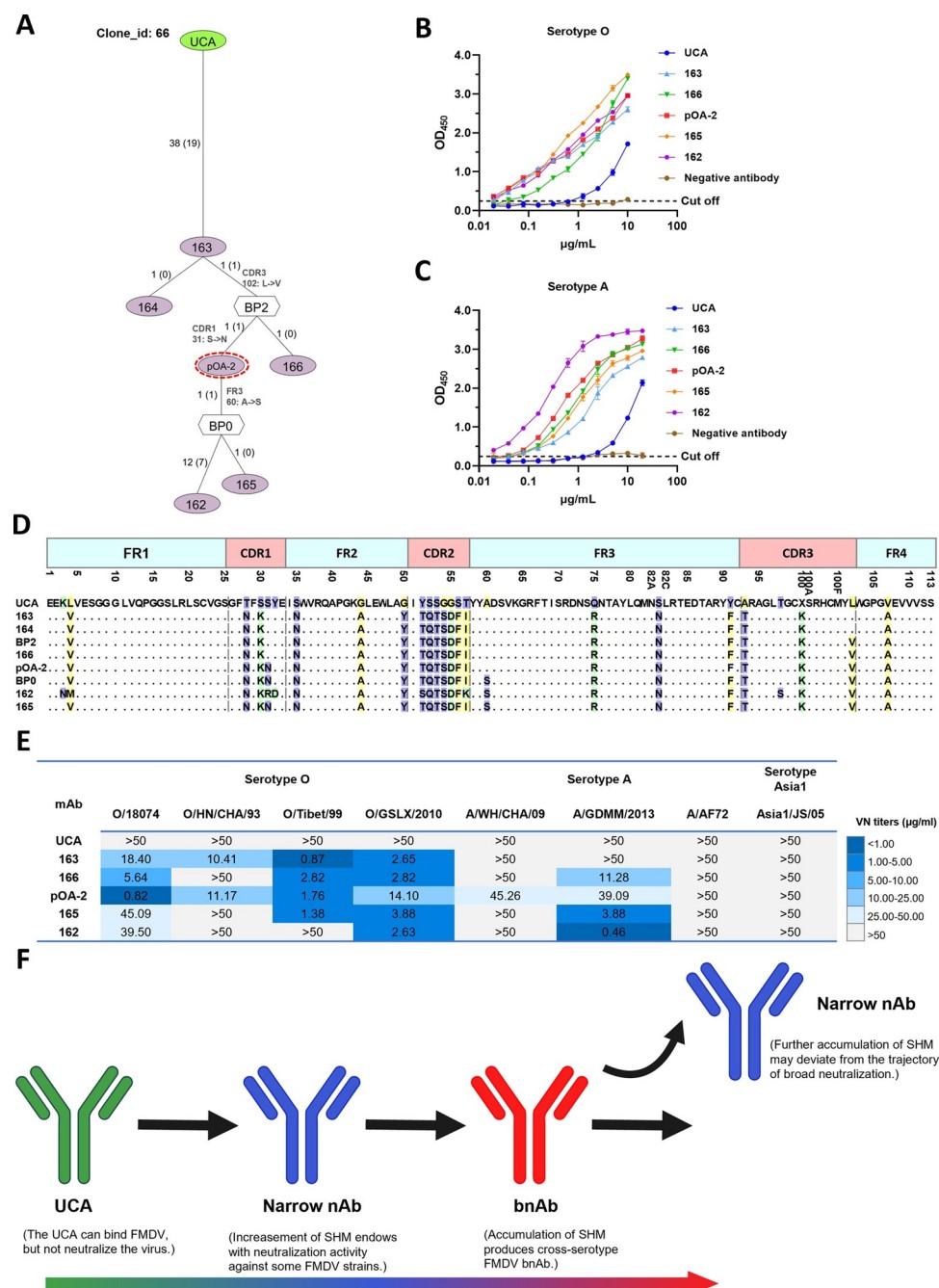

**Fig 8. The evolutionary pathway and occurring SHM of the porcine mAbs from the pOA-2-containing clonal lineage. (A)** Phylogenetic tree of pOA-2-containing clonal lineage (Clone 66), and the locations of each SHM occurred between two nodes/sequences were detailly shown in the right region of panel. **(B, C)** The binding ability of the porcine mAbs and their related UCA to FMDV serotypes O (O/Tibet/99) and A(A/AF72) antigens evaluated by indirect ELISA. **(D)** Alignment of VH amino-acid sequences of six antibodies in clone 66 with their UCA and branchpoint configurations. **(E)** Neutralization breadth and potency of porcine mAbs and UCA against representative strains of FMDV serotypes O, A and Asia1 tested by micro-neutralization assay. **(F)** The potential role of SHM in the evolution of cross-serotype bnAbs against FMDV. Created in BioRender. Chen, Y. (2023) BioRender.com/m35r709.

## FMDV bnAbs employ dual cross-serotype neutralization mechanisms via blocking attachment to cells and destabilizing the viral particle

To reveal the FMDV cross-serotype neutralization mechanism, three representative porcine-derived bnAbs (pOA-1, pOA-2 and pOA-13) were selected according to their difference in antigen recognition. The pOA-1 and pOA-13 recognized the linear VP1 G-H loop with distinct functional antigen determinations at position RGD+2 and +1 respectively, whereas the pOA-2 targeted a conformational epitope at the viral three-fold axis. Binding kinetics of the three bnAbs to serotypes O and A were measured by biolayer interferometry (BLI). As shown in Fig 9A and 9B, the pOA-2 and pOA-13 have similar affinities at nM level to both serotypes O (O/Tibet/99) and A (A/WH/CHA/09) 146S antigens. However, for binding kinetics of pOA-1, we observed extremely higher affinity to O/Tibet/99 (0.04 nM) strain versus A/WH/CHA/09 strain (2.48 nM), exhibiting slow association and dissociation for O/Tibet/99, but fast association and dissociation for A/WH/CHA/09. The dissociation rate of virus-antibody complex seems to correlate with efficacy of neutralization [32,33]. The discrepancy in pOA-1's neutralization efficiency to the O/Tibet/99 and A/WH/CHA/09 strains could be explained by this affinity kinetic characteristic. Considering the conservation of RGDL motif, which is bound by pOA-1, on the linear G-H loop epitopes of both strains, the slower dissociation rate observed for O/Tibet/99 compared to A/WH/CHA/09 may also indicate a disparity in the conformation of the flexible G-H loop on viral particles between serotypes O and A.

Attachment of cells is the first step for FMDV infection, and then the virus enters the cytoplasm of the host cell and release its genome. Generally, blocking cell attachment has been recognized as a major strategy for neutralization antibody. Thus, inhibition FMDV binding BHK-21 cells by porcine bnAbs were separately tested at two stages of pre- and post-attachment. For pre-attachment stage, the serotype O FMDV was first incubated with bnAb pOA-1, pOA-2 and pOA-13 at different doses, then the non-neutralized virus was allowed to attach BHK-21 cells. As shown in Fig 9, we observed the significantly reduced immunofluorescence signal (Fig K, panel A in S1 Appendix), structure protein (Fig 9C), viral RNA (Fig 9E), and plaque formation (Fig 9G) in the BHK-21 cells after incubation with pOA-1, pOA-2 and pOA-13, comparing with that for the FMDV negative porcine monoclonal antibody (PLY55). The consistent results were also observed for antibodies incubation with serotype A FMDV (Figs 9D, 9F and 9H, and K, panel B in S1 Appendix). Additionally, structure comparisons of FMDV-integrin and FMDV-antibody show obvious clashes between pOA-2 and the integrin receptor, suggesting that FMDV neutralization by the bnAb is facilitated by blocking virus-receptor interaction via steric hindrance (Fig L in S1 Appendix). These results showed that all three bnAbs are capable of blocking virus attachment to cells. For post-attachment assay, the bnAbs were incubated with the BHK-21 cells pre-absorption with FMDV and tested whether affecting viral entry. Specifically, pOA-2 also inhibited FMDV entry of BHK-21 cells in a dose dependent way, as evidenced by a combination of results shown in Fig M in S1 Appendix; pOA-1 and pOA-13 did not obviously affect viral entry of cells, only exhibiting partial viral RNA reduce in a condition of high antibody concentration. However, the observed results could result from the employed post-attachment assay that allowed incubation of antibody with the pre-absorbed cells with viruses at 37°C for 1 h. This protocol inadvertently allowed for the reactivation of endocytic machinery, enabling adsorbed viruses to undergo endocytosis before the mAbs could fully opsonize bound virions. Taking these factors into consideration, it is evident that the observed minimal post-attachment inhibition by pOA-2 is certainly real, but the way this assay was performed may well have underestimated the post-attachment neutralization this mAb is capable of. And the lack of post-attachment neutralization seen for the other mAbs may simply have been missed due to the way the assay was performed.

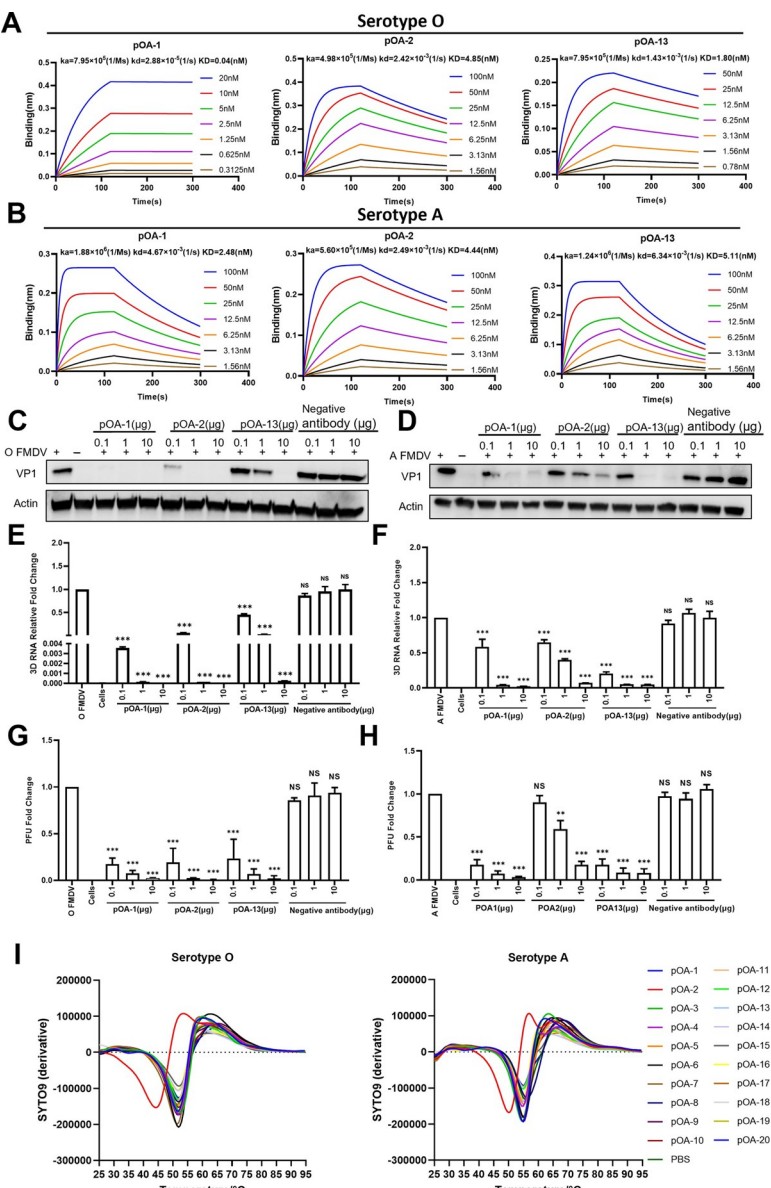

**Fig 9. Identification of cross-serotype neutralization mechanisms of porcine bnAbs against FMDV serotypes O and A. (A, B)** Binding affinities of porcine bnAbs (pOA-1, pOA-2 and pOA-13) to the inactivated FMDV 146S antigens of serotype O (O/Tibet/99 strain) **(A)** and serotype A (A/AF72 strain) **(B)** by biolayer interferometry on an Red96 system. The Ka, Kd and KD values are shown above the individual plots. **(C, H)** The inhibition effect of porcine bnAbs on viral attachment to BHK-21 cells. Different amounts of pOA-1, pOA-2 or pOA-13 were respectively pre-incubated with FMDV serotype O (O/HN/CHA/93 strain) **(C, E, G)** or serotype A (A/WH/CHA/09 strain) **(D, F, H)**. The cell-attached viruses were quantified through detecting VP1 protein by WB **(C, D)**, and 3D gene by qRT-PCR **(E, F)** and the numbers of PFUs on BHK-21 cells **(G, H)**. The experiments were independently conducted in triplicate. The data differences between conditions with virus only and different bnAb treatments were assessed using unpaired T-test (Holm-Sidak method, α = 0.05) in GraphPad Prism 7.2. *** Indicates an extremely significant difference at P<0.001. NS indicates no significant difference. **(I)** Thermostability assay of serotypes O and A FMDV treated with 20 porcine-derived mAbs. The Y-axis indicates the derivate of raw fluorescence representing the level of viral RNA release. PBS was used as a blank control.

Next, we determined whether porcine bnAbs neutralize FMDV via an alternative mechanism, such as destabilization of virion structure and initiation of viral RNA release or more-violent mechanism by dissociation of viral particle. We investigated their effect on the stability

of FMDV upon binding with all the obtained 20 antibodies by PaSTRy assays using SYTO9 dye to detect exposure of viral RNA. Distinct with the other antibodies and PBS control, binding of pOA-2 clearly decreases the stability of FMDV particles, with $T_R$ being reduced by 5°C. The consistent results were observed for FMDV serotypes O and A (Fig 9I). Thus, bnAb pOA-2 employed a dual mechanism via blocking virus attachment to cells and destabilization of viral particle, to perform cross-serotype FMDV neutralization.

## Discussion

This study initially reported the protocol for construction and dissection of porcine antigen-specific BCR repertoires that enables the rapid discovery of cross-serotype bnAbs against FMDV. We determined two distinct cross-serotype neutralizing antigenic sites existing on FMDV serotypes O, A and Asia1(Fig N in S1 Appendix). The predominant cross-serotype antigenic site on G-H loop of VP1 properly overlaps with viral receptor recognition site, which could mediate the most efficient neutralization mechanism via direct blocking virus attachment. Another cross-serotype antigenic structure mapped on FMDV three-fold axis and covered the residues distributing on two adjacent pentamers, which may mediate antibody neutralization by destabilizing of viral particle. Given to the high variation and multiple serotypes of FMDV, these cross-serotype viral neutralization mechanisms may represent a universal strategy of bnAb for other members within the *Picornaviridae* family. Antigenic structure mapping on the inter-pentamer interface has also been demonstrated in other *Picornaviruses* via the utilization of neutralizing antibodies such as A9 (anti-enterovirus 71) and R10 (anti-hepatitis A virus), with its involvement in the maintenance of viral particle stability [34–36].

The RGD-recognizing integrins are expressed on fundamentally different cell types to fulfill the most diverse biological tasks, thus self-antigen would be the RGD triplet acting as a widespread cell recognition motif in a conformation similar to that found in FMDV [37,38]. Elimination of self-reactive cells could inhibit the repertoire of neutralizing antibodies directed to structures which contain an RGD triplet. However, successful obtainment of the "RGDL"-recognizing bnAbs from porcine BCR repertoires suggests that B cell precursor can undergo self-tolerance to evolve into plasma cell secreting bnAbs. Accumulation of higher SHM is associated with more neutralization breadth and potency, as frequently observed for human bnAbs against SARS-CoV-2, HIV and influenza virus [39–42]. Porcine bnAbs binding the VP1 G-H loop of FMDV are primarily derived from terminally differentiated B cells with high accumulation of SHM. Conversely, the evolution of bnAbs targeting a different antigenic structure at the three-fold axis of viral capsids may unveil a distinct role of SHM in enhancing antibody neutralization breadth. The acquisition of an optimal range of SHM is crucial for improving neutralization breadth among B cell members within the same clonal lineage and recognizing the identical epitope.

There are several limitations in the immunogenetic analysis of porcine BCR repertoires. Firstly, due to under-sampled and under-sequenced germline Ig gene segments in the utilized reference library, allelic variation in Ig genes among individual pigs in the same species can lead to an overestimation of SHM levels. Some abnormally high SHMs of porcine B cells observed in this study could be attributed to the under-sampling of germline Ig gene segments in pig. Secondly, during the classification of clonal lineages and construction of evolutionary trees using the Immcantation program, certain BCRs failed to obtain their corresponding clonal group or evolutionary tree, such as bnAbs pOA-19 and pOA-13. This phenomenon may arise from incomplete representation of porcine germline Ig gene segments in the reference library used. Lastly, it is important to note that data on antibody repertoires were derived solely from one pig immunized with FMDV antigens. However, whether this response is

unconventional or is indicative of a typical response in pigs will not be known until multiple pig repertoires are characterized.

The protuberant G-H loop of VP1 on FMDV was shown to be immunodominant, as revealed by many reports of strain or serotype specific mAbs against type O, A, C and Asia1. Our study confirmed that the "RGDL" motif in G-H loop of VP1 protein is contained in the inter-serotype conserved epitopes, enabling it to induce bnAbs response in pig. The coincidence with FMDV receptor recognition site enables these antibodies to effectively recognize all potential variants, as viral escape from receptor recognition results in their inability to infect cells and subsequent demise. A dual participation of capsid amino acids in receptor recognition and antibody binding has recently been observed also for influenza viruses [12] and sarbecovirus [43]. This unveils a universal antiviral mechanism for bnAbs through mimicking receptor recognition. To date, the conformation of the VP1 G-H loop on the surface of FMDV particles remains elusive. However, speculation arises from diverse available monoclonal antibodies targeting this site, suggesting that the exposed VP1 G-H loop may exhibit mobility while its internal conformation is strain or serotype-dependent, regulated by specific residues after the RGD triplet, particularly at the RGD+1/+2 positions. The core motif "RGDL" crucial for recognition by both viral receptor and bnAbs may play a pivotal role in supporting the conformation of the G-H loop.

Dissection of the cross-serotype antigen structures pioneers a new way to explore the universal FMDV vaccine. Early study showed that synthesized tandem-"RGDL" peptides invoked neutralizing IgM antibodies with low titers but a broad specific for different FMDV serotypes and variants in vaccinated rabbits and guinea pigs, however repeated vaccination significantly decreased the antibody response [44]. FMDV VP1 G-H loop peptides as vaccine was intensively studied by different groups and was designed as different formats, showing part or complete protection in immunized bovine and pigs [45–47]. However, recently, the synthesized VP1 G-H loop peptides vaccine was withdrawn due to lack of enough protection in clinical application. This suggests the "RGDL" motif was not simply a linear epitope, and exhibition of full immunogenicity similar to that of G-H loop on viral particle needs suitable carrier for maintaining the correct conformation of "RGDL" motif, which can be recognized by bnAbs-producing B cells lineages. The utilization of anti-idiotypic antibody for mimicking the viral antigenic structure requires reassessment and shows considerable potential as a mean to develop a broad-spectrum antigen[48–50]. Currently, our ongoing study reveals that anti-idiotypic antibodies targeting pOA-2 exhibit significant immunogenicity and can bind with UCA. Hence, integrating a germline-targeted approach into the design of a universal FMDV antigen vaccine remains an active area of investigation within this field, with ongoing studies underway.

## Materials and methods

### Ethics statement

All the animal experiments in the present study were approved by the Review Board of Lanzhou Veterinary Research Institute, Chinese Academy of Agricultural Sciences (Permit No. LVRIAEC2021-006) and conducted in accordance with the Animal Ethics Procedures and Guidelines of the People's Republic of China on animal use.

### Virus production, inactivation and purification of 146S antigens

FMDV (O/HN/CHA/93, O/18074, O/Tibet/99, A/WH/CHA/09, or A/AF72) was propagated in the baby hamster kidney cells (BHK-21) that have been cultured adherently in MEM medium using a 5-layer cell factory system (NEST, China) at 37°C with 5% $CO_2$ for 48 h. After

incubation with FMDV under the same conditions for 12 h, the cell supernatant was collected following three freeze-thaw cycles. Viruses in supernatant were inactivated by treating with 1.2% Binary ethylenimine (BEI) at 30°C for 28 h, followed by the addition of a stop solution containing 4% sodium thiosulfate to neutralize the effect of BEI. The deactivated viruses were checked in BHK-21 cells for three successive passages, and no CPE was observed within each passage for a duration of 48 h, indicating complete viral inactivation. For purification of 146S antigen, approximately 1 liter of the inactivated viruses were precipitated by incubating at 4°C overnight in 8% (w/v) PEG 6,000. The precipitated virus antigens were harvested by centrifugation at 3,500 g for 1 h at 4°C, followed by resuspension in 50 ml PBS (137 mM NaCl, 2.7 mM KCl, 50 mM $Na_2HPO4$ and 10 mM $KH_2PO_4$, pH = 7.6). Subsequently, viral antigens were pelleted through a cushion of 30% (w/v) sucrose in PBS by centrifugation at 35,000×g for 4 h. The sucrose was removed from the pellet, and 500 µl of PBS was added to cover the pellet. The supernatant was further purified over a 20–60% sucrose gradient and fractionated by centrifugation at 35,000 g for 4 h at 4°C.The fractions were analyzed by negative stain electron microscopy, and the fraction corresponding to 146S antigen was transferred to a 100 kDa MWCO centrifugal filter for buffer exchange with PBS to remove the sucrose. The concentrations of the 146S antigens were quantified by a spectrophotometer at 260 nm (where an optical density of 7.6 = 1 mg/ml), and immediately used for subsequent experiments.

## Porcine vaccination

Three-month old healthy Hezuo pig (Sus scrofa), a miniature and primitive species, was raised in a clean animal room and primarily received two immunizations at one month interval with inactivated vaccine of FMDV serotype O (O/HN/CHA/93 strain) (5 µg 146S antigen formulated with ISA201 adjuvant). On day 157 after first immunization, the pig was further boosted with bivalent vaccine of serotypes A and O containing four strains of the viral antigen, O/18074 (Cathay topotype strain isolated from diseased pigs in 2018 from Zhengzhou, Henan province in central China), O/Tibet/99 (ME-SA topotype), A/WH/CHA/09 (G1 subset of SEA97 lineage) and A/AF72 (A22 lineage). The dose of corresponding 146S antigens were 5 µg each formulated with ISA201 adjuvant. On the 5[th] day after the final vaccination, approximately 250 ml of the EDTA anticoagulation blood was sampled from peripheral of pig and layered on HISTOPAQUE 1.077 (Sigma-Aldrich, USA) for isolation of PBMCs by centrifugation at 600×g for 30 mins.

## Purification of FMDV-specific porcine B cells from PBMCs

Approximately $1×10^9$ PBMCs suspended in 5 ml complete RPMI 1640 medium were incubated with a combination of mouse anti-porcine mAbs (30 µg each) including anti-CD3, anti-CD14, anti-CD335 and anti-IgM (Bio-Rad, USA) for 30 mins on ice. After two washes, the PBMCs were further incubated with 2 ml anti-mouse IgG microbeads for 30 mins on ice. After washing, these PBMCs were loaded onto an LD column for negative separation and the effluent cells were collected for staining and subsequent FACS. Briefly, the highly purified FMDV 146S antigens were first biotinylated with NHS-LC-Biotin reagent (Thermo Fisher Scientific, USA) according to the manufacturer's instructions, after that the resulting biotin-146S antigens (O/18074 strain and A/AF72 strain) were respectively used as bait to bind specific B cells, followed by incubated with anti-biotin APC antibody. Finally, these stained cells were loaded on FACSAria II (BD) for sorting O/18074 (serotype O) -binding B cells and A/AF72 (serotype A) -binding B cells, respectively.

## High-throughput sequencing of porcine BCR and transcripts of single B cells

After purification by FACS, the O/18074-binding and A/AF72-binding single cells were resuspended in PBS buffer. Immediately, single-cell capturing and library construction were performed using the Chromium Next GEM Single Cell 5' Kit v2 (10x Genomics) (PN-1000263) with a target loading of 10,000 cells per reaction according to the manufacturer's instructions. Briefly, the cell suspension, barcoded gel beads and partitioning oil were loaded onto the 10x Genomics Chromium Chip K to generate single-cell Gel Beads-in-Emulsion (GEMs). Captured cells were lysed and the transcripts were barcoded through reverse transcription inside individual GEMs. Then cDNA along with cell barcodes were PCR-amplified. The scRNA-seq libraries were constructed by using the 5' Library Kits (PN-1000190), and the scBCR-seq libraries were constructed by using the Human B Cell V(D)J Enrichment Kits (PN-1000252), where primers for amplification of V(D)J fragments were replaced with the customed porcine BCR primers panel (Table A in S1 appendix). Each sample was processed independently, and no cell hashing was applied. The constructed libraries were sequenced on an Illumina NovaSeq 6000 platform to generate 2×150-bp paired-end reads.

## Porcine BCR repertoires analysis

BCR sequences were assembled and quantified following Cell Ranger (v.4.0.0) vdj protocol against porcine V(D)J reference that was generated by the command-line mkvdjref using the IMGT/V-QUEST reference (*Sus scrofa*) as input. Assembled contigs labeled as low-confidence, non-productive or with UMIs < 2 were discarded. The resulting filtered_contig.fasta and filtered_contig_annotations.csv were reanalyzed using R (v4.3.1) software to produce paired Clonotype that was defined as B cell clone containing exactly paired heavy and light chains as well as identical combination of HCDR3+LCDR3 amino acid sequence. The frequency of Clonotype was determined by counting the number of distinct cell barcodes for each unique HCDR3+LCDR3. Those cells in clones supported by only one cell were considered as unexpanded clones (singleton), whereas those clones supported by two or more cells were considered as expanded.

The clonal lineages and SHM of porcine antibodies were analyzed by the program Immcantation. We aligned the paired BCR contig to IMGT reference genes using HighV-Quest [51] and the output was parsed using Change-O and the SHM was calculated by SHazaM tool [52]. The SHM was calculated based on mutations per nucleotides in the entire variable region sequence (VDJ region) of immunoglobulin heavy chain (VH). The frequency of SHM was showed in percentage (% nt SHM). For very few mAbs, their SHMs were calculated based on mutations per nucleotides in the V gene segment only, because of the failure in identifying the sequences which clones they belong to. The evolutionary tree of bnAbs-containing clonal lineage was constructed by IgPhyML [30,31] and visualized in the AcesTree [53].

## *In vitro* expression of porcine mAbs

The selected pair of VH and VL genes were codon-optimized and respectively cloned into porcine heavy chain (CH-pcDNA3.4) and light chain (Cκ-pcDNA3.4 or Cλ-pcDNA3.4) expression vectors to obtain the whole antibody-expressing plasmids VH-CH-pcDNA3.4 and VL-Cκ-pcDNA3.4/or VL-Cλ-pcDNA3.4, as described in our previous report [54]. The recombinant formatted single-chain fragment variable (scFv) was designed by joining VH and VL fragments using a flexible linker (GGGGSGGGGSGGGGS) and a His tag (HHHHHH) added at the C-terminus. The final optimized scFv gene was cloned into pcDNA3.4 expression vector.

The scFv or the antibody expressing plasmids with a light-to-heavy chain ratio of 3:2 was transfected into the suspended 293F cells (Invitrogen, USA) and continued cultivation for 5 days. The expressed mAbs in supernatants were initially purified through Ni-chelating affinity chromatography and further purified by size exclusion chromatography using a Superdex 200 increase 10/300 column in an AKTA plus protein purification system (GE Life Sciences). The concentration of expressed mAbs was determined by measuring the absorbance values at a wavelength of 280 nm (A280).

## Virus neutralization test

The porcine mAbs and serum samples were titrated for viral neutralizing activity against multiple FMDV serotypes O, A and Asia1 strains, including O/18074, O/HN/CHA/93 strains (Cathay topotype), O/Tibet/99 strain (ME-SA topotype) and O/GSLX/2010 strain (SEA topotype), A/AF72 strain (A22 lineage), A/WH/CHA/09 and A/GDMM/2013 strains (SEA97 lineage) in Asia topotype and Asia1/JS/05 strain, as well as the rescued virus by using a micro-neutralization assay as previously described [55]. Briefly, antibody samples were 2-fold serially diluted in 96-well cell culture plates in a total volume of 50 μl, and then 100 $TCID_{50}$ of FMDV in 50 μl of culture media was added to each well. After incubation for 1 h at 37°C, $\sim 5 \times 10^4$ BHK-21 cells in 100 μl medium were added to each well as indicators of residual infectivity. Normal cell wells, 0.1, 1, 10 and 100 $TCID_{50}$ virus control wells in duplicate were used in each plate. The plates were incubated at 37°C under 5% $CO_2$ conditions for 48 h before observing cytopathic effect (CPE). The experimental results were acceptable when complete CPE and no CPE appeared separately in 0.1 $TCID_{50}$ and 100 $TCID_{50}$ virus control wells. The endpoint titers were calculated as the reciprocal of the last serum dilution to neutralize 100 $TCID_{50}$ FMDV in 50% of the wells. Neutralizing activity is expressed as the VN titer, which was calculated as the initial antibody concentration divided by the endpoint titer.

## Cryo-EM sample preparation, data collection and three-dimensional reconstruction

FMDV O/18074 146S and A/WH/CHA/09 146S were individually incubated with scFv at a molar ratio of 1:240 in a volume of 10 μl for 10 to 60 mins at 4°C. A 4-μl aliquot of the mixture was applied to a glow-discharged Quantifoil grid (Au 1.2/1.3, 200 mesh). The grid was blotted for 3 s in 100% relative humidity and plunge-frozen in liquid ethane using a Vitrobot mark IV (Thermo Fisher, USA). Cryo-EM data were collected at 300 kV with a Titan Krios G3i (Thermo Fisher, USA) and a direct electron detector (K3 Bioquantum, Gatan) at Lanzhou University. Micrograph images were collected as movies (15 frames, 1.5 s) and recorded at −2.0 to −0.8 μm under focus at a calibrated magnification of ×105 kX, resulting in a pixel size of 0.83 Å.

The raw micrograph movies were imported into cryoSPARC[56] for processing and three-dimensional (3D) reconstruction. Individual frames from each micrograph movie were aligned and averaged to produce drift-corrected images. Particles were automatically picked and selected, and contrast transfer function (CTF) parameters were estimated using CTFFIND4 [57]. Subsequent steps in 3D reconstruction, including ab-initio, heterogeneous, and homogeneous reconstructions, were all carried out in cryoSPARC. For all reconstructions, the resolution was assessed using the standard FSC = 0.143 criterion. The data collection and refinement statistics are summarized in Table I in S1 appendix.

## Model building and data analysis

To construct an initial model of FMDV-O18-POA2, the cryo-EM structure of native FMDV-OTi (PDB:7D3K) [15] and the predicted structure of pOA-2 by using AlphaFold2 [58]

were manually placed into the cryo-EM map for FMDV-O18-POA2 complex and rigid-body fitted with UCSF Chimera[59]. The complex model was further improved with real-space refinement using Phenix [60]. Manual model building was performed using Coot [61] in combination with real-space refinement with Phenix to adjust mismatches between the model and the density map. The FMDV-AWH-POA2 model was built and refined using the similar procedure as above, except that the virus structure in FMDV-AWH-R50 (PDB:7D3R) [15] was used as the starting model of FMDV. The final models were validated by using MolProbity [62].

## Selection of neutralization-escape mutants using porcine mAbs

Neutralization escape mutants were generated by consecutive passages of FMDV in BHK-21 cells under the pressure of neutralizing mAbs according to a previous report, with minor modifications [63]. The representative FMDV strain (O/HN/CHA/93) was employed to select mutants against these mAbs. Briefly, 10-fold serial dilutions of FMDV in 50 µl were incubated with 50 µl of various concentrations of mAbs (20 µg/ml to 50 µg/ml) in 96-well microplates. The mixtures were used to infect 100 µl of BHK-21 cells ($10^6$ cells/ml), which were incubated at 37°C for 48 h to allow virus propagation occurrence. First-passage viruses were harvested from wells seeded with the highest dilution of virus that produced an approximately 80 to 100% CPE. Subsequent rounds of pressure selection were performed in 24-well plates, in which the passaged virus (200 µl) was incubated with an equal volume of a 2-fold concentration of antibodies in each well containing 400 µl of BHK-21 cells. The harvested virus was subjected to several rounds of selection until it completely escaped neutralization after addition of mAbs at concentrations of at least 400 µg/ml. The P1 region of the obtained neutralization escape mutants was amplified by one-step reverse transcription-PCR (RT-PCR), as described previously [64], using the primer pair Pan2041 (ACCTCCAACGGGTGGTACGC)/NK61 (GACATGTCCTCTTG-CATCTG) and subsequently verified by sequencing. Mutated amino acids were determined by aligning the entire mutant P1 region to the sequence of its initial parent virus.

## Rescue of site-directed FMDV mutants by reverse genetics

Full-length cDNAs of FMDV serotype O were generated using an existing pOFS plasmid that contained the entire P1 gene of O/HN/CHA/93. Full-length cDNAs of FMDV serotype A were generated using an existing pQQN plasmid with the entire P1 gene of A/WH/CHA/09. Site-directed mutagenesis was applied to introduce nucleic acid mutations to produce full-length cDNAs with single amino-acid substitutions [65]. All mutant constructs were confirmed by nucleotide sequencing. The site-directed FMDV mutant viruses were rescued as previously described [66]. Briefly, *Not* I-linearized mutant plasmids were transfected into BSR/T7 cells using Lipofectamine 2000 according to the manufacturer's instructions. The transfected cells were monitored daily for CPE appearance. At 72 h post-transfection, the cells were harvested and passaged in BHK-21 cells. After 3 rounds of passaging, the mutant virus titers were determined in BHK cells by calculating the 50% tissue culture infectious dose ($TCID_{50}$), subsequently used to perform micro-neutralization assay as described above.

## Bio-layer interferometry (BLI) assay

Before BLI assay, purified O/18074 and A/WH/CHA/09 viral particles were respectively labeled with Sulfo-NHS-LC-LC-Biotin (Thermo Fisher Scientific, USA) following manufacturer's instructions and then purified using spin desalting column (Thermo Fisher Scientific) to remove excess non-reacted biotin. To determine binding affinity of the bnAbs to FMDV (serotype O and serotype A), BLI assay was performed in an RED96 System according to manufacturer's instructions. Briefly, biotinylated O/18074 and A/WH/CHA/09 virion (10 µg/ml) was

respectively immobilized onto streptavidin-coated biosensors (Pall ForteBio) until saturation. The antigen-bound biosensors were placed in wells containing a series of diluted bnAbs to allow antigen-antibody association and were then dipped into dissociation buffer (0.01 M PBS supplemented with 0.1% bovine serum albumin and 0.02% Tween 20). The on-rate (ka) and off-rate (kd) were determined by global fitting of the association and dissociation phases of a series of bnAbs concentrations. The equilibrium dissociation constant (KD), a measure for affinity, was then calculated as the ratio of kd and ka.

## Inhibition of virus attachment assay

Different amounts of mAbs were mixed with $5\times10^5$ TCID$_{50}$s of FMDV (O/HN/CHA/93 strain or A/WH/CHA/09 strain) in MEM medium, followed by incubation at 37˚C for 1h. The mixtures were loaded on pre-cooling BHK-21 cells (approximately $5\times10^5$ cells per treatment) and incubated at 4˚C for 1h to allow virus attachment. After three thorough washes with cold PBS to remove unbound virus, the cells were further cultured for a duration of 4 h to enhance quantitation of the virus, then the final cells were harvested and examined by immunofluorescence staining, WB, plaque forming unit (PFU), and qRT-PCR assays. For the post-attachment inhibition assay, the BHK-21 cells were pre-adsorbed with equal amount of virus, and then were incubated with different amounts of mAbs at 37˚C for 1h. After washing with cold PBS to remove unbound virus, the cells were further cultured for a duration of 4 h, subsequently the samples were collected and analyzed as below. Immunofluorescence staining was performed in 24-well plate, and added to cattle-derived FMDV-specific antibody (E32) 200 μl/well at a concentration of 5 μg/ml for 1h at 37˚C, followed by incubation with rabbit anti-cattle FITC-IgG (Thermo-Fisher, USA) at a 1:1,000 dilution for 1h at 37˚C. Plates were washed three times with PBS and observed under an FL Imaging System (Life Technology, USA). WB was performed to 12% sodium dodecyl sulfate-polyacrylamide gel electrophoresis (SDS-PAGE) and electro-transferred to a methanol-activated PVDF membrane, and blocking with 5% non-fat milk in PBS overnight at 4˚C. The membrane was probed with porcine-derived mAb (pOA-20) diluted to 2 μg/ml and rabbit anti-β-actin polyclonal antibody at a 1:2000 dilution. The expression levels of indicated proteins were normalized with GAPDH. The qRT-PCR assays were performed with a pair of FMDV-3D-gene specific primers (forward primer, ACTGGGTTTTAYAAACCTGTGATG; reverse primer, TCAACTTCTCCTGKATGGT CCCA), and a pair of β-actin-specific primers (forward primer: TGACGTGCCGCCTGGAG AAA; reverse primer: AGTGTAGCCCAAGATGCCCTTCAG). The relative expressions of FMDV-3D gene were normalized to β-actin. Data analysis was performed using the $2^{-\triangle\triangle Ct}$ method relative to the control group. The threshold cycle (C$_T$) values of all samples were first normalized to the C$_T$ value of β-actin and then compared to the C$_T$ value of the control (samples treated with virus only). The inhibition efficiency of virus attachment for a given treatment group was defined as the FMDV-3D gene copy number for the group as a percentage of that for the control (which was treated with the virus only).

The plaque forming unit (PFU) assay was performed in 6-well plate using tragacanth gum (0.6%). Plaques were visualized for 48h at 37˚C by fixing with acetone-methanol and staining with crystal violet. The amount of plaque was observed and statistically analyzed. The inhibition efficiency of virus attachment for a given treatment group was defined as the PFU number for the group as a percentage of that for the control (which was treated with the virus only).

## Thermofluor assay

Thermofluor experiments were performed using a RT-PCR instrument (ABI, Thermo Fisher Scientific) to evaluate FMDV 146S particle stability after incubation with each mAb. In thin-

walled PCR plates (ABI, Thermo Fisher Scientific), a 50-µl reaction volume was established using mixtures of 1.0 µg of 146S particles plus 1.5 µg antibody ($\sim$60 antibody molecules per FMDV virion) and 5 µM SYTO9 (Invitrogen, USA). For all assays, the melt temperature was set from 25˚C to 95˚C in 0.5-˚C increments with intervals of 1s. Fluorescence was evaluated with excitation and emission wavelengths of 490 nm and 516 nm, respectively. The dynamics of viral RNA release from virions with temperature were detected by increases in the fluorescence signal. Three independent assays were performed for each analysis. Data sets exported from the PCR machine were visualized using GraphPad Prism 7.2.

## Supporting information

**S1 Appendix. Fig A. Sorting of FMDV-specific B cells using different bait antigens by fluorescence-activated cell sorting (FACS). (A)** Distribution and proportion of FMDV O serotype (O/18074) of specific B cells in the porcine peripheral blood mononuclear cells (PBMCs) identified by flow cytometry. Gate 1 (P1) was selected to exclude cell debris, with lower values of SSC-A and FSC-A, and further analyzed singlets in gate P2 based on diagonal streak of the FSC-A and FSC-H plot. The class-switched (IgG[+]) B cells in gate P3 were used to check the distribution of FMDV-specific cells, in the presence of FMDV serotype O (O/18074) bait antigen. Appropriately one million counts were collected to analyze the proportion of O/18074-binding B cells in PBMCs. **(B)** Sorting of the FMDV A serotype (A/AF72) or FMDV O serotype (O/18074) specific B from the enriched B cells population using FACS. After excluding cell debris to gate singlets in P2, the class-switched B cells being IgM[-]CD14[-]CD3[-]CD335[-] population were gated in P3 to sort the FMDV-specific cells in the presence of bait antigen A/AF72 and O/18074, respectively. **(C)** Determination of the purity of the sorted FMDV-specific B cells by flow cytometry. After sorting, the purified B cells were reloaded and one thousand counts were collected to check the proportion of O/18074-binding B cells. **Fig B. The phenotype and constitution of FMDV-binding cells revealed by scRNA-seq transcripts. (A)** UMAP plot of unsupervised clustering of FMDV-binding cells, comprising of majority B cells, as well as other minimal cell populations such as monocytes, T cell, dendritic cells (DCs) and unidentified cells. **(B)** Heatmap of differentially expressed genes in B cells, Monocytes, T cells, DCc, and others. **Fig C. The heterogeneity of porcine memory B cells revealed by pairing analysis of BCR and transcripts. (A)** Heatmap of differentially expressed genes between memory B cells and PBs. **(B-F)** UMAP plot of unsupervised clustering of porcine memory B cells, comprising of six clusters in **B**. According to the germline gene usages of VL, the memory B cells were separated into kappa V1-expressing B cells (pairing with cluster 3 in **C**), kappa V2-expressing B cells (pairing with cluster 2 in **D**), lambda V3-expressing B cells (pairing with cluster 4 in **E**) and lambda V8-expressing B cells (pairing with clusters 0, 1 and 5 in **F**). **Fig D. Analysis of the relative abundance and density of BCR clonotypes for serotypes O and A specific BCR repertoires.** (A) The proportion of BCR clonotypes was shown in O/18074-specific and A/AF72-specific repertoires respectively. (B) The number of the clonotypes with different frequency indicated the difference in abundance between the two repertoires. **Fig E. Analysis of the SHMs of porcine BCR repertoires. (A)** The SHM difference of porcine VH between memory B cells and plasmablasts in FMDV serotype O-specific repertoire. **(B)** The SHM difference of porcine VH between memory B cells and plasmablasts in FMDV serotype A-specific repertoire. **(C-F)** The difference in SHM of each isotype antibodies between the serotypes O and A specific BCR repertoires. The statistical analysis was performed using nonparametric Mann-Whitney test in R program and showed the medians in violin plot. **(G)** The difference in SHM among the isotypes (IgA, IgD, IgE and IgG) antibodies within each serotypes O and A specific BCR repertoire. The statistical analysis was performed using Kruskal-

Wallis chi-square test in R program and showed the medians in violin plot. P<0.05 indicates a significant difference between two samples. P<0.01 indicates a very significant difference between two samples. P<0.001 indicates an extremely significant difference between two samples. NS indicates no significant difference. Fig F. Identification of the reactivity of porcine mAbs with FMDV serotypes O and A using indirect immunofluorescence assay (IFA). (A, B) BHK-21 cells infected respectively with the O/18074 strain (A) or A/AF72 strain (B), and the working concentration of the tested porcine mAbs was 5μg/ml, followed by incubation with rabbit anti-pig FITC (diluted 1:200 in PBS). The cells were observed under an FL Imaging System (Life Technology, USA). The experiments were independently conducted in triplicate. Fig G. Identification of the reactivity of porcine mAbs with FMDV serotypes O and A using enzyme-linked immunosorbent assay (ELISA). (A, B) The ELISA plates were respectively coated with inactivated 146S antigen of O/Tibet/99 (A) or A/AF72 (B), and then probed with different concentrations of 0–40 μg/ml of the tested mAbs, followed by probing with HRP-conjugated goat anti-porcine IgG. Color was developed by adding 50 μl of TMB substrate (Pierce, Life Technology) for 10 min at room temperature. The process was stopped by adding equal volumes of 1 M $H_2SO_4$. Optical density at 405 nm ($OD_{450}$) was measured on a microplate reader (BioRad). The experiments were independently conducted in triplicate. Fig H. Analysis of sequence conservation of VP1 G-H loop of available FMDV strains in serotypes O, A and Asia1. The full VP1 amino acids sequences of available FMDV serotypes O, A and Asia1 were downloaded from national center for biotechnology information (NCBI) as of June 30, 2023. The sequence logo of VP1 G-H loop of FMDV serotypes O (numbers of full VP1 sequences = 6317), A (numbers of full VP1 sequences = 2352) and Asia1 (numbers of full VP1 sequences = 812). The key antigenic determinants on G-H loop that recognized by porcine bnAbs were marked with bold black line. The integrin receptor (αvβ6) recognition motif, "RGD", was framed with rectangles and marked with yellow. Fig I. The resolution of the cryo-EM reconstruction complex. (A, B) Fourier shell correlation (FSC) of FMDV-O18-POA2 complex (A) or FMDV-AWH-POA2 complex (B). Fig J. Identification of the rescued single-substitution mutants by plaque formation assay. (A) The wild-type (O/18074) and rescued mutants (VP2 D68A, VP2 T71A, VP2 N72A, VP2 R77A, VP2 P195A, VP3 D69A, VP3 S70A and VP3 D195A) formed in BHK-21 cells, and the sizes were correlated to the CPE patterns. (B) The wild-type (A/WH/CHA/09 strain) and rescued mutants (VP2 H65A, VP2 D68A, VP2 T71A, VP2 P195A, VP3 S70A) formed in BHK-21 cells, and the sizes were correlated to the CPE patterns. Fig K. Neutralization mechanism of porcine bnAbs against FMDV serotypes O and A. The inhibition effect of porcine bnAbs on viral attachment to BHK-21 cells was determined by IFA. Different amounts of pOA-1, pOA-2 or pOA-13 were respectively mixed with FMDV serotype O (O/HN/CHA/93 strain) (A) or serotype A (A/WH/CHA/09 strain) (B) at 37˚C for 1 h, and the mixtures were loaded on BHK-21 cells at 4˚C for 1 h to allow virus attachment, then washed 3 times with cold PBS to remove unbound virus. The viruses were detected by IFA. The experiments were independently conducted in triplicate. Fig L. Binding modes of FMDV integrin receptor and antibody. Binding modes of FMDV integrin receptor (avβ6) and bnAb pOA-2. (A) Superposition of FMDV- avβ6 with FMDV-O18-POA2 (A) and FMDV-AWH-POA2 (B). VP1, VP2, VP3 and VP4 of the protomer are shown in blue, green, red and yellow, respectively. The av and β6 chains of integrin (avβ6) and pOA-2 are drawn in cartoon representation and colored in cyan, limon and magenta, respectively. Black dashed circles show significant clashes between antibody (pOA-2) and integrin receptor. Fig M. Effect of porcine bnAbs on virus at the post-attachment stage. The BHK-21 cells were respectively incubated with FMDV serotype O (O/HN/CHA/93 strain) (A, C, E, G) or serotype A (A/WH/CHA/09 strain) (B, D, F, H) at 4˚C for 1 h. Subsequently, the cells were treated with different amounts of pOA-1, pOA-2 or pOA-13 at 37˚C for 1h, the cells were washed with cold PBS to

remove unbound virus, then the cells were further cultured for a duration of 4h. The viruses were quantified through detecting VP1 protein by IFA (A, B) and Western blotting (C, D), 3D gene by qRT-PCR (E, F) and the numbers of PFUs by plaque phenotypes assay (G, H). The experiments were independently conducted in triplicate. The data differences between conditions with virus only and different bnAb treatments were assessed using unpaired T-test (Holm-Sidak method, α = 0.05) in GraphPad Prism 7.2. *** Indicates an extremely significant difference at P<0.001. ** Indicates a very significant difference at P<0.01. * Indicates a significant difference at P<0.05. NS indicates no significant difference. Fig N. The recognized antigen structures of porcine cross-serotype bnAbs against FMDV. Footprints of two distinct cross-serotype antigenic sites on surface of two protomers from two adjacent pentamers of FMDV. One protomer comprising of VP1, VP2, VP3 and VP4 was circled in grey line. Cross-serotype antigenic site 1 that consisted of VP1 143, 145–148 and 151 position residues was marked in pink. Cross-serotype antigenic site 2 that consisted of VP2 65, 68, 71, 72, 77 and 195 position residues on one protomer and VP3 68, 69, 70 and 195 position residues on another protomer, was marked in orange. Table A. Porcine single B cell sequencing primers used for V (D)J amplifications. Table B. The interactive residues between the integrin receptor (alpha V beta 6) with VP1 G-H loop on FMDV. Table C. Porcine broad neutralizing mAb escape mutants. Table D. Interface identification and interaction analysis of pOA-2 with FMDV O/ 18074 by PISA Program. Table E. Interface identification and interaction analysis of pOA-2 with FMDV A/WH/CHA/09 by PISA Program. Table F. FMDV O/18074 with pOA-2 interaction residues. Table G. FMDV A/WH/CHA/09 with pOA-2 interaction residues. Table H. Sequence identity of pOA-2 epitopes in FMDV serotypes O and A. Table I. Cryo-EM data collection and refinement statistics.
(DOCX)

**S1 Table. The immunogenetics annotation of common clonotype B cells using Immcantation program.**
(XLSX)

**S2 Table. The sequences of all the porcine-derived monoclonal antibodies in this study.**
(XLSX)

## Acknowledgments

We thank the Supercomputing Center of Lanzhou University for providing facilities. We thank the staff at Instrument Centre, Lanzhou Veterinary Research Institute, Chinese Academy of Agricultural Sciences for excellent assistance in virion purification and cell sorting using BD FACS Aria II. We also thank Xiaoqi Wu (Genergy Biotechnology Shanghai Co., Ltd) for his help with the bioinformatics analysis and for generously sharing his experience and code.

## Author Contributions

**Conceptualization:** Fengjuan Li, Shanquan Wu, Zaixin Liu, Zengjun Lu, Dongsheng Lei, Kun Li.

**Data curation:** Fengjuan Li, Shanquan Wu, Huifang Bao.

**Formal analysis:** Fengjuan Li, Shanquan Wu, Shulun Huang, Zelin Zhang, Zhaxi Zerang, Pinghua Li.

**Funding acquisition:** Zengjun Lu, Dongsheng Lei, Kun Li.

**Investigation:** Fengjuan Li, Shanquan Wu, Lv Lv, Yong He, Yuanfang Fu, Hong Yuan, Xueqing Ma.

**Methodology:** Fengjuan Li, Shanquan Wu, Pinghua Li, Yimei Cao, Hong Yuan.

**Project administration:** Zaixin Liu, Zengjun Lu.

**Resources:** Fengjuan Li, Shanquan Wu, Lv Lv, Shulun Huang, Pinghua Li, Yimei Cao, Huifang Bao, Xingwen Bai, Dong Li, Qiang Zhang, Jijun He.

**Software:** Shanquan Wu, Shulun Huang, Jian Wang, Tao Wang, Kun Li.

**Supervision:** Zaixin Liu, Dongsheng Lei, Kun Li.

**Validation:** Fengjuan Li, Pu Sun, Zhixun Zhao, Jing Zhang.

**Visualization:** Fengjuan Li, Shanquan Wu, Xingwen Bai, Yuanfang Fu, Kun Li.

**Writing – original draft:** Fengjuan Li, Shanquan Wu, Kun Li.

**Writing – review & editing:** Fengjuan Li, Shanquan Wu, Zengjun Lu, Dongsheng Lei, Kun Li.

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
