## [Decision Letter · Decision Letter 0]

6 May 2024

Dear Dr Li,

Thank you very much for submitting your manuscript "Discovery, evolution, and recognized antigenic structures of cross-serotype broadly neutralizing antibodies from porcine B-cell repertoires against foot-and-mouth disease virus" (PPATHOGENS-D-24-00480) for review by PLOS Pathogens. Your manuscript was fully evaluated at the editorial level and by independent peer reviewers. The reviewers acknowledged the considerable amount of work presented, and appreciated the attention to an important problem, but raised some substantial concerns about the manuscript as it currently stands. These issues must be addressed before we would be willing to consider a revised version of your study. We cannot, of course, promise publication at that time. We therefore ask you to modify the manuscript according to the review recommendations before we can consider your manuscript for acceptance. All three reviewers provided reasoned, precise and thorough feedback in their comments. Your revisions should address the specific recommendations made by each reviewer.

Collectively, concerns were expressed on the overall clarity and conciseness of the manuscript, the unintuitive organization of the data, imprecise language, and somehow lacking information to fully assess the rigor. Critically, serious concerns were raised on the annotation and clonal assignment of the B cell V(D)J sequences and the appropriateness of the statistical analysis applied to the repertoire-level analysis of SHM. Also, lack of statistical clarity throughout the manuscript and incomplete descriptions of the methods were mentioned.

Reviewers #1 and #2 emphasized that the large amount of data presented make the manuscript particularly dense. There were conflicting opinions on which sections may be de-emphasized or removed. I suspect that such discordance is mostly a reflection of suboptimal manuscript organization and writing, rather than data being not relevant. I strongly suggest proceeding first with addressing the critical concerns raised, then reorganizing the manuscript and parsing primary and supplementary data according to revised analyses, and finally streamline the text with particular attention to better explain the reasoning and significance of individual analyses (for example, the relevance of focusing on SHM, or IFA), if still deemed necessary.

We are returning your manuscript with three reviews. Based on the reviews and reading the manuscript, we recommend Major Revision. We are sorry we cannot be more positive at the moment; however, we are looking forward to receiving your revision. With a lot of work, the manuscript will be suitable for a resubmission, if you so wish to do so. Note that the revised manuscript will be sent out for further review upon resubmission.

Please, pay particular attention to the following suggestions from the reviewers and give them due consideration.

[1] Revise the immunogenetic annotations using rigorous programs (such as Partis, Immcantation, or alternative similar programs) to group clones appropriately and reconstruct the lineages for clones with more firmly established clonal relatedness between members. Given the centrality of clonal assignment in supporting major analyses and conclusions (e.g., clonal maturation, ancestry, and relationship between SHM and function), the authors should pay particular attention to validate their clonal grouping. Please include a separate table of the immunogenetics of the POA antibodies which establishes the V, D, J gene usage in the heavy and light chains, along with CDR3 lengths and SHM of light and heavy chains for easy reference.

[2] Revise/add details on statistical analysis performed (see reviewer #2 and #3 comments). Seeking advice from a statistician would be appropriate.

[3] Reorganize the manuscript to clarify figures/data that are part of the main story from less important ‘supplementary’ data. In particular, present the study design earlier in the manuscript, as suggested by reviewers #1 and #3.

[4] Revise text for accuracy and clarity, including abstract and methods (multiple comments from all reviewers, including IFA and ELISA methods from reviewer #2).

[5] State the limitations of the study in the discussion (see reviewer #3 comments).

[1] A letter containing a detailed point-by-point list of your responses to the review comments and a description of the changes you have made in the manuscript. Please note while forming your response, if your article is accepted, you may have the opportunity to make the peer review history publicly available. The record will include editor decision letters (with reviews) and your responses to reviewer comments. If eligible, we will contact you to opt in or out.

[LINK]

Additional information for resubmission are included below the reviewers' comments.

Sincerely,

Mattia Bonsignori, M.D., M.S.

Guest Editor

PLOS Pathogens

Ashley St. John

Section Editor

PLOS Pathogens

Kasturi Haldar

Editor-in-Chief

PLOS Pathogens

orcid.org/0000-0001-5065-158X

Grant McFadden

Editor-in-Chief

PLOS Pathogens

orcid.org/0000-0002-2556-3526

Reviewer's Responses to Questions

**Part I - Summary**

Reviewer #1: A single mini pig was immunised with a prime and boost of a single O serotype vaccine antigen. Approximately 4 months later, the pig was immunised with two different O serotype antigens and two serotype A antigens. Neutralising antibody titres generally developed at the expected times post immunisation (O-specific responses developed after immunisation with O antigens and A-specific responses only developed after the later immunisation with A antigen). Interestingly, after the immunisation with A antigens, a neutralising response also developed against a third serotype, Asia1.

B cells from PBMCs harvested after the final immunisation were enriched, separated by specificity to O or A serotype and B cell receptor repertoires characterised. Clonotypes shared between the O- and A-specific repertoires were identified, expressed as recombinant mabs and found to be reactive with O and A and also with Asia1.

Somatic hypermutation (SHM) and clonal evolution was investigated for several antibody lineages containing broadly neutralising antibody clonotypes. From one lineage containing broadly neutralising antibody pOA-2, the authors propose a model where the most broadly neutralising antibodies may be lost due to further SHM which selects for increasing potency of neutralisation at the expense of breadth.

The structures of mab pOA-2 interacting with FMDV serotypes O and A are solved showing the interaction is across two pentamers. The antibody interaction is slightly different between O and A serotypes but with some conserved features, especially the involvement of VP2 D68 which was found to be critically required for ab binding and which is very highly conserved across O, A , Asia1 serotypes.

The other nine broadly neutralising antibodies characterised were all found to react with a linear epitope which mapped to the GH loop, with the RGDL integrin-binding motif forming the core of the epitope. The GH loop residues involved in interactions with integrin are proposed to be the same as those targeted by the broadly neutralising mabs.

Generation of escape mutants against the antibodies and single site viral reverse genetics showed that one or both of RGD +1 and RGD+2 appeared to be the most important residues required for antibody neutralisation of both serotypes. For three mabs, the kinetics of binding to virus was determined and the ability of mabs to prevent attachment to cells or entry into cells was investigated (some of this questionable). Mab pOA-2 appears to destabilise the virus.

This is a lot of work with some interesting findings and should certainly be published.

In my opinion this paper includes too much data and could have been broken up into at least two separate papers! Supplementary material should supplement the main results (e.g. provide additional detail) but should not be required to understand the main story. In this paper, the reader is constantly referred to the supplementary material and it is often not clear whether this is part of the main story or a less important ‘supplementary’ part of the story. I may not have reviewed in detail all of the supplementary material.

I found some parts of the text in the results section associated with figures 1 and 2 were confusing and could perhaps be written more clearly. I have given some examples and suggestions for extra clarity throughout the manuscript in the minor issues section.

For some of the work investigating frequency of SHM I felt the findings and their significance could be better explained. However, the results section associated with fig 3 seemed clearly explained and was a strong point.

My biggest recommendation for this paper is to explain the simple/conventional parts of the study first (i.e. immunisation schedule currently Fig 2D and the time course of serotype specific polyclonal response currently Fig 2E). This would make everything else much easier to understand.

The structural biology seemed convincing to me and the discovery of a bnAb able to bind across pentamers of multiple serotypes with a destabilising mechanism is interesting and a strong point of the study.

The GH loop is immunodominant and presumably it was therefore expected to identify many GH reactive antibodies? However I was surprised that so many RGD binding antibodies could be cross reactive between serotypes. Was this expected? How do these findings sit with existing knowledge of GH loop reactive monoclonals? Is this a special feature of broadly neutralising mabs?

The binding studies and IF are weak. There is so much data in this paper, is this even required?

Reviewer #2: Foot-and-mouth disease virus (FMDV) is a member of the Picornaviridae family that causes severe disease and death in cloven-hoofed animals, including domestic cattle, pigs, sheep, and goats. There are seven FMDV serotypes which do not induce cross-protection in infected animals, and there is need for cross-protective vaccines. This group and others have laid some foundation for this already when they discovered antibodies that are cross-reactive and neutralize Serotype O and A FMDV in previous publications. However, these antibodies were derived from a bovine and camelid source, which have uncharacteristically long heavy chain CDR3 loops or single chain antibodies, respectively. It remained to be seen whether antibodies with more traditional, flatter binding surfaces can also cross-react with multiple FMDV serotypes.

Here, the authors develop a method for discovering monoclonal antibodies that cross-react to multiple FMDV serotypes. The developed method sorts single B cells from FMDV vaccinated pigs using whole virus particles to stain B cells, then uses single-cell RNA-sequencing to determine the sequence of the most abundant FMDV-specific B cells. Through this process, the authors sorted thousands of FMDV-reactive B cells, of which 216 bound to both serotype O and A viruses, and they discovered 10 monoclonal antibodies that neutralize all tested strains of FMDV serotypes O, A (except A/AF72), and Asia1. While most of the B cells from which they derive these antibodies were terminally differentiated, one (pOA-2) was not and its progeny lost the ability to neutralize some strains following additional somatic hypermutation. The authors determined that 9/10 of these cross-neutralizing antibodies recognized a highly conserved RGDL motif in FMDV VP1 G-H loop, which is known to be important for binding the virus’ integrin receptor. The final cross-neutralizing antibodies (again, pOA-2) bridges protomers and pentamers in its interactions with VP2 and VP3. Thermofluor assays indicate that this inter-pentamer binding antibody, but not the RGDL motif-binding antibodies, can cause pre-mature release of the viral genome at physiological temperature.

Reviewer #3: In their manuscript, “Discovery, evolution, and recognized antigenic structures of cross serotype broadly neutralizing antibodies from porcine B-cell repertoires against foot-and-mouth disease virus,” Li et al. describe work characterizing the antibody response to FMDV vaccination. The authors immunize a single pig primed with inactivated FMDV (O serotype) and boosted with a bivalent vaccine (A and O serotypes), sort B cells 7 days post-boost for FMDV reactivity and perform B cell receptor repertoire profiling via single B cell sequencing with the 10x Genomics platform on the sorted B cells. The authors devise a clever strategy for isolating cross-serotype antibodies by using the singe B cell sequencing platform to identify shared clones between two independent B cells sorts, on each of the O and A serotypes. The authors then go on to characterize several of these monoclonal antibodies for binding, neutralization capacity, immunogenetics and, for one antibody POA-2, structure. All broad neutralizing antibodies except POA-2 bind to a linear epitope in the G-H loop of VP1 with antibody recognition focused on a RGDL motif that the authors show is conserved across FMDV serotypes owing to its role in receptor binding. The POA-2 structure revealed that it bound to the VP2/VP3 interface another area of cross-serotype conservation. Overall, the manuscript represents an impressive amount of work characterizing the porcine antibody response to FMDV vaccination, however the impact of their findings is tempered by limitations in how the immunogenetic analysis was performed and the manuscript is not helped by the authors’ unpolished writing and general sloppiness in the presentation of the data.

 **********

**Part II – Major Issues: Key Experiments Required for Acceptance**

Reviewer #1: N/A

Reviewer #2: (No Response)

Reviewer #3: On lines 199-201, the authors state, “…we grouped BCR clones with identical V gene usage and heavy chain CDR3 length into the same lineage”. Grouping B cells into clones based solely on identical V gene usage and heavy chain CDR3 length is inappropriate. Membership in a B cell clone is defined as shared evolutionary descent from a common ancestor with a unique V(D)J rearrangement. V gene usage and heavy chain CDR3 length alone does not define a unique V(D)J rearrangement. There are immunogenetic annotation programs like Partis or Immcantation (which the authors use here for SHM analysis and thus should be familiar with) that can do clonal grouping properly which utilize the CDRH3 sequence to better establish common ancestry. The authors must use those (or alternative programs) to group clones appropriately and reconstruct the lineages for clones with more firmly established clonal relatedness between members.

The authors should include a separate table of the immunogenetics of the POA antibodies which establishes the V,D, J gene usage in the heavy and light chains, along with CDR3 lengths and SHM of light and heavy chains. While some of this information is scattered about in Table 1 and S2, it could be presented more clearly and fully in its own table. If the authors had done that, they may have recognized that 8 out of their 9 linear epitope cross-serotype neutralizing antibodies utilize the porcine VK2-10 light chain which seems like an important finding here given the convergence in recognition by those antibodies for the VP1 G-H loop epitope.

A major limitation in any mutation analysis of antibodies from an animal species with under-sampled and under-sequenced germline Ig gene segments is that allelic variation in Ig genes between individual animals within the same species can result in overestimation of SHM levels. What appears as a mutation is actually a difference between the sequenced pig’s alleles and the germline Ig gene segments in the library used. The mutation frequencies reported for these BCR repertoires and monocolonal antibodies seem quite high for a 9-month old pig. This could be due to under-sampled germline Ig gene segments in pig. The authors should discuss this as a study limitation.

I also found the repertoire-level SHM analysis to lack the appropriate measures and statistical tests given how SHM data is distributed. SHM is not normally-distributed, as is very evident from the violin plots included in Figure 1. However, the authors reported means instead of medians, and applied the Games-Howell statistical test, which assumes normality, when they should have used the Mann-Whitney test. The authors interpretation that in panel 1F plasmablasts are higher in SHM than memory B cells may not hold if they use the appropriate comparison of the medians. Furthermore, is the difference between 0.08 and 0.09 really even biologically meaningful, especially in light of the limitations based on pig allelic variation discussed above? Rather, it would seem the more relevant finding is that the memory B cells mutation frequencies cover a wider range than plasmablasts, and include some unmutated B cells as well as B cells with extremely high mutation frequencies (>20%) whereas the plasmablasts have no unmutated B cells and also no B cells with >20% mutation frequency.

The authors conclude from Figure 1G that based on differences of SHM between the O and A-specific populations, that “high affinity IgG antibodies against O/18074 strain may be biasedly produced in the vaccinated pig,” but isn’t the more likely explanation that the A response has simply not had time to mature given the sequencing was performed on samples from only 7 days after the pig was immunized with A (by bivalent boost) whereas the pig had 164 days of maturation time from the original priming immunization with O only?

The entire results sub-section “Somatic hypermutations balance the breadth and potency of neutralization” misuses the term “clones” and should use the term “members” instead throughout when describing antibodies that are related by evolutionary descent from an unmutated common ancestor (UCA). Again, the conclusions in Figure 3 are tempered by the way the clonal grouping was performed and must be validated by a program such as Partis or Immcantation.

I found the ordering of the data presentation to be odd. It would seem to be more logical to show the epitope mapping in Fig 6 establishing that 9 of the 10 shared clones map to linear epitopes, and that only pOA-2 binds to a conformational epitope, prior to showing the pOA-2 cryo-EM structures in Figures 4 and 5. As another example, the vaccine regimen is not shown until Figure 2D. It would be better if this was shown as the first panel in Figure 1 and the vaccine regimen described (including the prime and boost and interval) at the beginning of the Results section to orient the reader at the outset.

The manuscript would be strengthened by careful editing of the writing. There were various places where the wrong word was used and contributed to the overall impression of a lack of attention to detail by the authors in the preparation of the manuscript. For example, the second word in the abstract is “excavation”, but surely the authors mean “isolated” since they did not extract the antibodies from the dirt. Line 285: “Promoter” should be “Protomer”. Line 977: “Recused” should be “Rescued”.

 **********

**Part III – Minor Issues: Editorial and Data Presentation Modifications**

Reviewer #1: Polite request to the authors: Any suggestions, questions, or requests I make are intended to improve the manuscript, ultimately for the benefit of those reading the final published paper. If I suggest that additional information might be useful, my intention is for the extra information to be incorporated into the manuscript (not just provided to me as a reviewer in the rebuttal). Therefore wherever possible, please respond by adding information to the manuscript.

Abstract

The abstract is poorly written and confusing and currently does not do justice to the work of this study

line 23 “excavation” wrong word

line 23-25 poor language linking components of the sentence

The description of serotype specific responses/clonotypes/repertoires/antigens is confusing.

Line 31 could be interpreted to mean “all” antibodies targeted RGD, which conflicts with line 36 where the inter-pentamer epitope is introduced.

Line 37 “conservative determinant” what does this mean?

Author summary

Also some poor writing, please check/improve if possible.

I would advise against describing your own work as elegant. This is best left for commentary by others.

Introduction

61 Topotypes are not explained.

63 Suggest “…Asia1 is endemic in parts of Asia…”

67 “has been around” rather informal scientific language

69-82 make clear you are describing a general situation here (not just FMDV)

69-70 Suggest “immunologists endeavor to identify and characterize broadly neutralizing antibodies (bnAbs)”.

71 “within the body” ?

76 “recognition structures” ?

77 why is the reference to antiviral drugs relevant for this study?

79-82 not really sure what this is supposed to mean

84 “big” suggest ‘large’

84-87 not clear. please make clear what was the key finding from previous studies (by you and others) in the area of anti-FMDV B cell repertoires and broadly neutralising antibodies in large animals.

88 How can cattle antibodies be single chain molecules?

93 elegant (see previous comment)

93 mining (wrong word)

100 “recognized antigenic structures” words in wrong order

Results

119-120 confusing. .. if PCR amplification is required before sequencing it should be described in that order.

121-122 I don’t understand this sentence

126 suggest “87% of total isotypes (Fig 1B).”

131-133 why?

137 indicated the potential presence of conserved antigen structures… ??

138-139 not explained well… is any of this useful: ‘The top highest frequency clonotypes in both O and A repertoires also contained four of the clonotypes shared between O and A.’

Is this finding expected? I was surprised. If broadly reactive responses are rare, why are the shared clonotypes found at the top of the high frequency list? Is there some sort of comment that could be added to address this?

144-146 and Fig 1F I’m not expert on SHM. Can you explain somewhere in the results or figure legend the principle of how the SHM frequency values and Umean (strange symbol) are calculated. I’m guessing the SHM frequency is expressed as a something per something (perhaps this is in the methods). It would be useful to explain why the data in Fig 1F is considered to be ‘notably’ and ‘very significantly’ different, for example the plasmablast population appears to have a bi-modal distribution, with a sub-population of cells with higher SHM, is this what makes it so special? (for example the difference in SHM frequency between serotypes is even greater in Fig1G but this is not highlighted?).

Fig1G IgE is there enough data to be confident in these?

149-151 poor sentence

152 significantly increased relative to what? why does the text refer to a % value, this is not shown anywhere in the figure?

155-157 “…difference in humoral response…” what is this difference, can it be explained? “…high affinity ab against O may be biasedly produced…” why? what is the rationale for this statement. Is this referring to data which has not yet been explained?

162 “the top 10 clones with SHM and the top 10 high frequency clones” why were these types of clones selected, why were these considered to be the most likely to be bNabs out of the 216 clonotypes? e.g. link to the previous findings.

164 “biological activity” would a more specific phrase be more useful?

167 “binding affinity” where is the data on binding affinity?

168 “only minor variations…” this statement does not match the data in FigS5, some mabs clearly have less activity than others.

why is the data for O so different between IF and ELISA?

171-172 “can effectively reflect FMDV-specific humoral immune response” what does this mean?

Why is the IF and ELISA data described in the text as a primary part of the results but the data only shown as supplementary figures?

175-177 Virus strains include some not part of the immunisation strategy, how were these chosen?

Table 1 I do not find this a good way to summarise the VNT data. The colour coding is not intuitive and is not even explained with a colour key. I would recommend to find a better way to present this data.

How does the VNT data correlate with the IF and ELISA data?

191-194 I don’t understand this sentence.

202-203 “ten bnAbs were involved in six clonal lineages…” what does this mean? is involved the best word to use here?

206-209 states “majority of these bnAbs…” but Fig2 only shows 4 antibodies out of 10 (not a majority).

211 “significantly impacted the breadth of cross-serotype neutralization against FMDV” the evidence for this is not clear from Fig 2C. Is the next sentence intended to provide the evidence for this? If so I don’t understand.

214 “evolutionary character” what does this mean?

214-222 Why is this not shown earlier? Showing this earlier in the paper would make everything much easier to understand…

223-233 Apologies I became confused here. Why was bnAb pOA-20 chosen to be used here? There is mention of pigs with repeat immunisation with A antigen but this did not happen in this study? The conclusion of this section needs further explanation.

266-267 The location on the virus capsid of a novel virus-antibody interaction which is broadly neutralising via an unusual destabilising mechanism. This sounds like essential information for this study, I don’t understand why it is relegated to supplementary information!

272 “promoters” protomers

286-287 sentence doesn’t make sense

341 “concretely” not the best way to start a sentence

341-345 This confused me, where is this shown?

346-347 the RGD motif is expected to be highly conserved

350 “protrude” protruding

361-363 consider adding amino acid abbreviations for clarity e.g. 143N, 145R, 147D.

385-387 could remind here how these are different

Fif 7A and B identical panels, it would be useful to indicate on the figure which panel is which serotype

390-394 This seems important and could be explained more carefully . One ab has very striking slow dissociation, is this within normal expected range?

392-394 If this is referring to previous data in this study please make this clear.

Virus attachment assays: please make clear this is a direct assay of virus attachment at cell surface. Fig S11A does not look like surface labelling and is weak data.

Fig S13 what is this IF supposed be showing, virus at surface or new virus after replication? Legend says entry has been allowed at 37C then mabs added??

Reviewer #2: Overall, the data presented in this manuscript are convincing and support the claim that the authors have discovered several porcine monoclonal antibodies that neutralize multiple FMDV serotypes via distinct mechanisms. The various lines of biochemical evidence additionally highlight the importance of specific residues in the mAb binding sites that are critical for cross-neutralization. However, there is much description of somatic hypermutation (SHM) frequency in Fig 1 that could be moved to supplemental data, as the figures are quite dense overall and these data seem to only describe the intuitive phenomena of having more SHM in response to multiple exposures to a particular antigen (O strain FMDV) and less SHM in response to fewer exposures to a particular antigen (A strain FMDV). While no additional experiments are required for publication, there are some comments for improvement and clarification.

Big picture:

1. The authors frequently refer to the discovered mAbs that neutralize strains from multiple serotypes as “broadly” neutralizing antibodies. Given that none of the discovered antibodies neutralizes all strains that were tested (including one that was included in the inoculation series), and that there were four FMDV serotypes that were not tested, it seems more appropriate to refer to these antibodies as “cross”-neutralizing.

2. The vaccination strategy and usage of pig(s) is difficult to understand. As best I can tell, all mAbs were isolated from a single pig as detailed in Fig 1. However, the data in Fig 2E may be from a single pig or multiple pigs, and Fig 2F-H must be from multiple pigs, right? What might help is putting the vaccine strategy for the monoclonal antibody source pig (which I understand to be similar to or identical to Fig 2D) as the very first figure of the manuscript. Then if this does indeed differ for the later panels of Fig 2, a separate schematic can be shown there. Similarly, pay very close attention to the use of the singular “pig” and plural “pigs” throughout the text.

3. There are no details of the statistical tests used except for in Fig 1. The rigor of the analysis is therefore hard to evaluate. This needs to be specified for all usages of statistical analysis.

Specific text:

The methods for the competitive ELISAs in Fig 2F-H are not described and need to be included.

Fig. S9 shows plaques from rescued escape mutants. While this is good evidence that these viruses do still replicate, it does not tell anything about the extent to which their replication is inhibited (other than differences in plaque size). Data on escape mutant replication kinetics, if already acquired, would be a nice addition and would speak to the ability of the virus to naturally evade cross-neutralizing antibodies.

Line 74: the authors state “Moreover, within the immune milieu, antibodies not only neutralize the pathogens but also trigger their mutations . . .”. This should be phrased differently. Antibodies do not trigger mutations, they select for viruses with mutations that enable escape from antibody recognition.

Line 155: “These information in SHM of the two repertoires indirectly reflected the difference in humoral response to the FMDV strains, suggesting the high affinity IgG antibodies against O/18074 strain may be biasedly produced in the vaccinated pig.” I think the authors are suggesting that there is a qualitative difference in how the vaccinated pig responded to the O strain, but really they just vaccinated three times with O antigens and only one time with A antigen. In other words, this is not a quality of the pig’s B cells, that they respond better to O antigens than A antigens. This is simply a byproduct of the chosen vaccine regimen. Of course the response to an antigen seen many times should be better than to an antigen seen fewer times.

Line 448: the authors state “Conversely, the evolution of bnAbs targeting a different antigenic structure at the three-fold axis of viral capsids may unveil a distinct role of SHM in enhancing antibody neutralization breadth. The acquisition of an optimal range of SHM is crucial for improving neutralization breadth.” The data presented here surrounding the relationship between SHM and FMDV neutralization breadth are ambiguous. The RGDL-motif cross-neutralizing mAbs are terminally differentiated, but additional SHM reduced neutralization breadth for the pOA-2 lineage. Additionally, in Table 1, multiple mAbs with SHM frequency similar to that for the cross-neutralizing antibodies did not neutralize any of the strains tested. It is certainly true that an optimal range of SHM is important for neutralization, but to correlate this with recognition of a given epitope seems unfounded.

Line 480: the authors state “Presently, anti idiotypic antibodies that target pOA-2 demonstrate significant immunogenicity and can bind with UCA.” Where is this data? If this is published elsewhere, it should be referenced. If it is unpublished data that is not shown, the authors should make this clear.

For the pre- and post-attachment mechanisms analyzed in Fig 7 and S13, the authors describe doing all of their analyses immediately within an hour of adsorbing virus to cells. Yet they have impressive fluorescence in their IFA and bands in their wetern blots. This seems more in line with allowing some amount of viral replication after the adsorption of virus. I would like to confirm whether or not the authors allowed some amount of hours or days for the virus to replicate after adsorption to cells.

Reviewer #3: In Figure 1E, “NA” is reported for two clones’ CDRH3s, but no explanation as to what “NA” means is given in the Figure 1 legend. Surely “NA” is not the amino acid sequence of the CDRH3 for those clones.

The two histograms in Figure S4 cannot be distinguished as they are shown. The authors should just show lines without a color fill to help the reader distinguish them. Figure S4 legend is missing a description.

Line 493 says “On 5th day after the final vaccination…blood was sampled”. However, Figure 2D seems to suggest blood sampling occurred 7 days after the final vaccination (immunization at day 157 and sampled at day 164). The authors should clarify this inconsistency.

The authors should be commended for the level of depth that they went to in order to characterize the antibody repertoire of an FMDV immunized pig. However, whether this response is unconventional or is indicative of a typical response in pigs will not be known until multiple pig repertoires are characterized. The authors should note this as a study limitation.

The authors should adequately describe in the text their RGD+1 numbering convention before using it.

Positive and negative control antibodies should be included for the neutralization assay data reported in panels 6I and 6J.

Figure 6 legend says “(I,I)” but should be presumably say “(I,J)”.

Figure 7 has two panels labeled “E”, presumably the one on the right should be labeled “F”.

The POA-13 antibody is missing from Table S2.

Line 512: “High-through sequencing” should be changed to “High-throughput sequencing”
---

## [Decision Letter · Decision Letter 1]

3 Sep 2024

Dear Dr Li,

Thank you very much for submitting your manuscript "Discovery, recognized antigenic structures, and evolution of cross-serotype broadly neutralizing antibodies from porcine B-cell repertoires against foot-and-mouth disease virus" for consideration at PLOS Pathogens. As with all papers reviewed by the journal, your manuscript was reviewed by members of the editorial board and by several independent reviewers. The reviewers appreciated the attention to an important topic. Based on the reviews, we are likely to accept this manuscript for publication, providing that you modify the manuscript according to the review recommendations.

The reviewers have positively received the revised version of the manuscript, which satisfactorily addressed prior queries. However, the additional information and clarifications provided in the revised version raised few additional but important questions that need to be addressed before considering the manuscript for publication. Please, see comments from Reviewer #2. Concerns should be addressed. Specifically, please focus on the following points:

- The reviewer raised two important issues relative to the pre- and post-attachment assays (see first and second paragraph in the "minor issues" section). Both of them can be addressed with rewording, without additional experimentation. For the second point, you may either rephrase the conclusions, opt to include a new set of experiments or, if you disagree with the reviewer's assessment, provide a detailed rebuttal.

- Sequences of the monoclonal antibodies subject of this body of work should be deposited in a publicly accessible database (e.g., GenBank), if not already deposited, and Accession numbers of the heavy and light chains variable regions should be reported in the manuscript.

- Please include methodology on virus production and (for the product used for vaccination) inactivation process.

Also, please specify on the y axis of Fig 1H if SHM refers to amino acids or nucleotides (aa or nt) and the units (e.g. % nt SHM). For Fig 2, please rephase the "% noted the SHM data of this mAb was calculated basis on V gene segment of heavy chain." in the legend for clarity. Ignore comment on Table S9 - I confirm it is included among the submitted files (for clarity: that CDR3 sequences are included in Table does is not substitute for submitting the whole V(D)J sequences) - and consider the minor text recommendations.

Sincerely,

Mattia Bonsignori, M.D., M.S.

Guest Editor

PLOS Pathogens

Ashley St. John

Section Editor

PLOS Pathogens

Michael Malim

Editor-in-Chief

PLOS Pathogens

orcid.org/0000-0002-7699-2064

Dear Dr. Li,

The reviewers have positively received the revised version of the manuscript, which satisfactorily addressed prior queries. However, the additional information and clarifications provided in the revised version raised few additional but important questions that need to be addressed before considering the manuscript for publication. Please, see comments from Reviewer #2. Concerns should be addressed. Specifically, please focus on the following points:

- The reviewer raised two important issues relative to the pre- and post-attachment assays (see first and second paragraph in the "minor issues" section). Both of them can be addressed with rewording, without additional experimentation. For the second point, you may either rephrase the conclusions, opt to include a new set of experiments or, if you disagree with the reviewer's assessment, provide a detailed rebuttal.

- Sequences of the monoclonal antibodies subject of this body of work should be deposited in a publicly accessible database (e.g., GenBank), if not already deposited, and Accession numbers of the heavy and light chains variable regions should be reported in the manuscript.

- Please include methodology on virus production and (for the product used for vaccination) inactivation process.

Also, please specify on the y axis of Fig 1H if SHM refers to amino acids or nucleotides (aa or nt) and the units (e.g. % nt SHM). For Fig 2, please rephase the "% noted the SHM data of this mAb was calculated basis on V gene segment of heavy chain." in the legend for clarity. Ignore comment on Table S9 - I confirm it is included among the submitted files (for clarity: that CDR3 sequences are included in Table does is not substitute for submitting the whole V(D)J sequences) - and consider the minor text recommendations.

Reviewer Comments (if any, and for reference):

Reviewer's Responses to Questions

**Part I - Summary**

Reviewer #1: Resubmission PPATHOGENS-D-24-00480R1 from Li and Wu et al.

Thank you to the authors for the additional work to modify and improve this manuscript.

Reviewer #2: Overall, I am pleased with the modifications made by the authors to the manuscript. They adequately addressed the minor concerns I had with the previous version, and the manuscript is nearly fit for publication. However, I have noticed that the conclusions drawn from the pre/post attachment experiments are not quite in-line with the way the experiments were performed. Rather than changing or doing new experiments, I think this can be addressed with some modification of the text. I have also included some minor textual recommendations that would improve the reading of the manuscript, were they to be incorporated.

Reviewer #3: The authors' revisions substantially have strengthened the manuscript. I recommend accepting the manuscript for publication.

**Part II – Major Issues: Key Experiments Required for Acceptance**

Reviewer #1: N/A

Reviewer #2: The authors need to include the entire variable region sequences of their mAbs. At the very least they need to include this for the 10 bnAbs that they focus on, but really any mAb mentioned in the manuscript should have the sequence published. This is field standard to allow for reproducibility for other groups. This could be in a separate supplementary Excel table as one possible suggestion.

Reviewer #3: None

**Part III – Minor Issues: Editorial and Data Presentation Modifications**

Reviewer #1: N/A

Reviewer #2: Regarding the pre- and post-attachment assays (shown in Figs. 9C-H and S10 and S12), the conclusion the authors draw from the data are not as definitive as they present them. First, for the pre-attachment assay in which the mAbs are mixed with virus prior to attachment to cells, it is certainly possible that the mAbs function by blocking the receptor as the authors conclude. However, it is also entirely possible to opsonize mAbs to a virion, then have that virion bind to an attachment factor or receptor, then have the mAbs block some step of entry or uncoating that comes after the attachment step. Therefore, the way to better interpret these data is that the mAbs are *capable* of blocking attachment to cells. But this is not definitive evidence of blocking attachment as the authors present it. While this may seem subtle, it’s an important distinction.

Second, regarding the post-attachment assays there is a fatal flaw that I now realize with the added detail provided around these methods, which is why I asked for them in the first review. The authors pre-adsorb virus to cells at 4°, but then they add antibody at 37° for an hour. In doing this, they allow for the endocytic machinery to start back up. Therefore, the viruses that adsorbed to the cell have a chance to endocytose before the mAbs have the chance to fully opsonize bound virions. Instead, the antibodies should have been allowed to opsonize the cell-adsorbed virions at 4° (which can be for longer than 1 hr, that’s not a problem). By never including a 37° incubation in the middle of the protocol, it is possible a stronger neutralization effect would have been detected. All of this considered, the correct interpretation of these data is that the small amount of post-attachment inhibition seen by pOA-2 is certainly real, but the way this assay was performed may well have underestimated the post-attachment neutralization this mAb is capable of. And the lack of post-attachment neutralization seen for the other mAbs may simply have been missed due to the way the assay was performed.

Nowhere in the Materials and Methods do the authors describe a) how they make their virus preps (specifically what are the cell lines used to make them, what are the purification steps), nor b) how the viruses are inactivated for the vaccines. Please include these details.

The y axis of Fig. 1H and the “SHM” column in Fig. 2 has no units. Frequency of SHM based on what calculation? Mutations per amino acid in a given sequence? If you could explain this metric to the reader a little better, it would help a lot as you reference SHM rate often during the manuscript, including being the basis for the entire paragraph of lines 154-173. It seems clear that larger numbers mean more mutations, but knowing the details of how this is calculated with an extra sentence or two in the methods would be helpful.

Table S9 is missing. I’m guessing this might have the sequences of the mAbs.

The panels Fig 4C and D would be fine to move to the supplement.

Minor text recommendations:

Line 69: remove “most” as part of “most noticeable”. No need for a superlative here.

Line 72: what exactly is meant by “diverse viral variations”? Different viruses? Mutants within specific groups of viruses? The word choice is vague and confusing.

Line 91: instead of “Y-shaped” the use of “bivalent” would be more appropriate.

Line 102: change wording to “the antigenic structures recognized by bnAbs reveal”

Line 103: change “the most direct” to “a direct”.

Line 120: “sophisticated” is subjective. Remove this word.

Line 149: “suggested” would be more appropriate than the term “indicated”, as the existence of conserved antigenic structure is only one possible interpretation of the data. I agree it’s a reasonable interpretation, but it is not the only one.

Line 170: “it all accumulated rather than high SHM in pig induced by two different strains”. This wording is confusing, please clarify.

Line 242: “The flexibility of G-H loop of VP1 on viral surface made the virus-receptor complex conformational heterogeneity”. This wording is confusing, please clarify.

Line 257: the authors use “drove the variations” when it is likely that they mean “selected for mutations”. Please clarify.

Line 979: in the description for panel 7A, specify why you use the different colors.

Reviewer #3: None

PLOS authors have the option to publish the peer review history of their article (what does this mean?). If published, this will include your full peer review and any attached files.

Reviewer #1: No

Reviewer #2: No

Reviewer #3: No

Figure Files:

Data Requirements:

Reproducibility:

References:

---

## [Editor Report · Decision Letter 2]

27 Sep 2024

Dear Dr Li,

We are pleased to inform you that your manuscript 'Discovery, recognized antigenic structures, and evolution of cross-serotype broadly neutralizing antibodies from porcine B-cell repertoires against foot-and-mouth disease virus' has been provisionally accepted for publication in PLOS Pathogens.

Best regards,

Mattia Bonsignori, M.D., M.S.

Guest Editor

PLOS Pathogens

Ashley St. John

Section Editor

PLOS Pathogens

Michael Malim

Editor-in-Chief

PLOS Pathogens

orcid.org/0000-0002-7699-2064

In the added text I found a typo ("basis" instead of "based" in line 590). It can be taken care of at proofing stage.

If possible, it would be great to add the composition of the medium used to grow the virus as well.

---

## [Editor Report · Acceptance letter]

9 Oct 2024

Dear Dr Li,

We are delighted to inform you that your manuscript, "Discovery, recognized antigenic structures, and evolution of cross-serotype broadly neutralizing antibodies from porcine B-cell repertoires against foot-and-mouth disease virus," has been formally accepted for publication in PLOS Pathogens.

Best regards,

Michael Malim

Editor-in-Chief

PLOS Pathogens

orcid.org/0000-0002-7699-2064